# Flood exposure and pregnancy loss in 33 developing countries

Cheng He [1,2,5], Yixiang Zhu[1,5], Lu Zhou[1], Jovine Bachwenkizi[3], Alexandra Schneider [2], Renjie Chen [1] ✉ & Haidong Kan [1,4] ✉

Floods have affected billions worldwide. Yet, the indirect health impacts of floods on vulnerable groups, particularly women in the developing world, remain underexplored. Here, we evaluated the risk of pregnancy loss for women exposed to floods. We analyzed 90,465 individual pregnancy loss records from 33 developing countries, cross-referencing each with spatial-temporal flood databases. We found that gestational flood exposure is associated with increased pregnancy loss with an odds ratio of 1.08 (95% confidence interval: 1.04 - 1.11). This risk is pronounced for women outside the peak reproductive age range (<21 or >35) or during the mid and late-stage of pregnancy. The risk escalated for women dependent on surface water, with lower income or education levels. We estimated that, over the 2010s, gestational flood events might be responsible for approximately 107,888 (CIs: 53,944 - 148,345) excess pregnancy losses annually across 33 developing countries. Notably, there is a consistent upward trend in annual excess pregnancy losses from 2010 to 2020, and was more prominent over Central America, the Caribbean, South America, and South Asia. Our findings underscore the disparities in maternal and child health aggravated by flood events in an evolving climate.

Catastrophic floods have intensified in severity, duration, and frequency due to recent shifts in climate, sea levels, infrastructure, and population dynamics[1–3]. Over the past few decades, floods have emerged as the most prevalent type of natural disaster[4], impacting 2.3 billion people and resulting in economic losses of over $ 600 billion globally[5–7]. The immediate aftermath of flood events are readily visible, including the destruction of physical environments, injuries, psychological stress, and infectious diseases[8]. Recent studies suggested the indirect health crisis triggered by flooding, as it compromises many ecological determinants, including safe drinking water, food security, and secure shelter[9,10]. The combination of these direct and indirect effects could disproportionately affect vulnerable communities. Previous studies have reported the impacts of flood exposure on some

specific diseases in the general population, such as psychological disorders[11], infectious disease[12], and gastrointestinal diseases[13]. Yet, there is a notable knowledge gap in the impacts of flood exposure on vulnerable groups, particularly pregnant women. These women could confront amplified challenges during floods, such as unsafe labor and delivery conditions[14], and exacerbated scarcity of essential resources like water and food[15,16]. In addition, floods may not only pose immediate health risks but also have far-reaching health impacts (e.g., mental illness) and enduring socioeconomic consequences. For example, pre-conception exposure to floods may significantly affect the health of pregnant women and the development of their fetuses[17]. Previous studies have indicated that maternal exposure to flood during pregnancy elevates the risk of pregnancy loss[18,19], a major public health

[1]School of Public Health, Key Lab of Public Health Safety of the Ministry of Education, NHC Key Lab of Health Technology Assessment, IRDR ICoE on Risk Interconnectivity and Governance on Weather/Climate Extremes Impact and Public Health, Fudan University, Shanghai, China. [2]Institute of Epidemiology, Helmholtz Zentrum München – German Research Center for Environmental Health (GmbH), Neuherberg, Germany. [3]Department of Environmental and Occupational Health, Muhimbili University of Health and Allied Sciences, Dar es Salaam, Tanzania. [4]Children's Hospital of Fudan University, National Center for Children's Health, Shanghai, China. [5]These authors contributed equally: Cheng He, Yixiang Zhu. ✉e-mail: chenrenjie@fudan.edu.cn; kanh@fudan.edu.cn

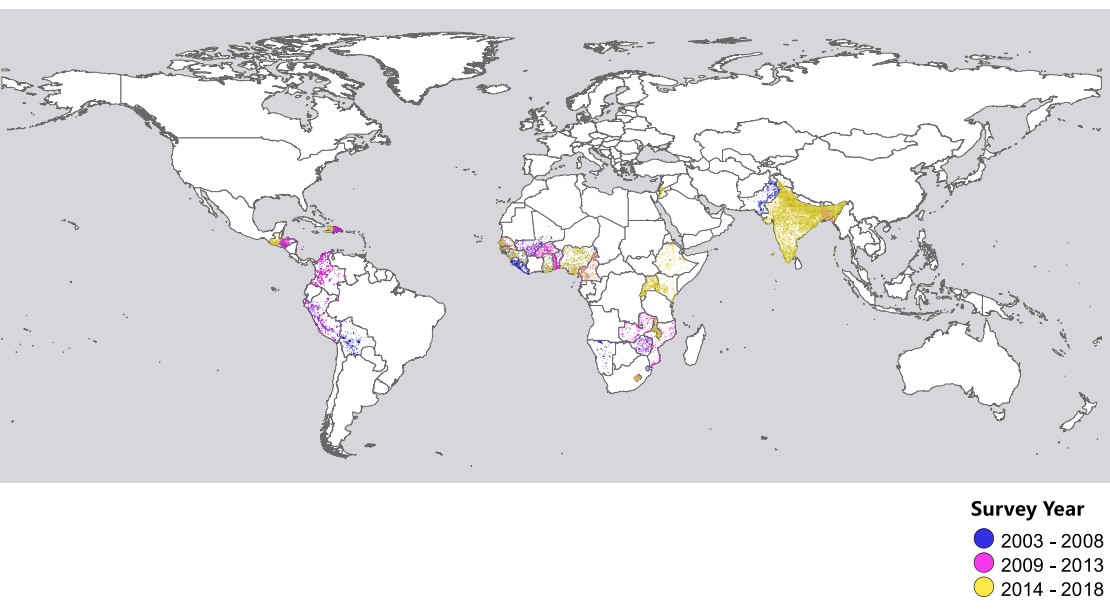

**Fig. 1 | These clusters include cases of women who have experienced both fruitful pregnancies and pregnancy losses and were exposed to floods during past pregnancies.** The spatial distribution of eligible clusters from 33 developing counties/regions, 2003–2018.

concern that is linked with substantial economic and emotional tolls[20]. However, prevailing epidemiological data linking maternal flood exposure to adverse birth outcomes often stems from small samples, isolated flood incidents, or confined regions[21,22], making it difficult to draw reliable conclusions.

Maternal and child health challenges related to floods are particularly acute in developing countries, as 89% of the global flood-exposed population reside in low- and middle-income countries (LIMICs)[5]. Furthermore, 75% of stillbirths worldwide are estimated to occur in developing countries within sub-Saharan Africa and South Asia[23]. The already-high prevalence of pregnancy loss in these regions could be compounded by the expansion of informal settlements and inadequate water and sanitation infrastructure[24]. Additionally, unfavorable socioeconomic circumstances and unsafe living conditions can diminish the resilience of pregnant women, rendering them particularly susceptible during flood events. This vulnerability arises at both the individual and social levels. At the individual level, limited healthcare access and elevated stress during flood events play a role; and at the social level, the absence of adequate infrastructure and support systems magnify the adversities pregnant women face during flood events[25]. Consequently, understanding the impact of flood exposure on pregnancy loss in developing countries holds significant importance for global public health. Such understanding offers invaluable guidance for effectively organizing the delivery of essential services and implementing targeted protective measures to safeguard vulnerable populations in the face of climate change. Nevertheless, studies on this topic, especially from a multi-region perspective, are rare. Additionally, the possible effect modifications by socioeconomic and living conditions are also not well understood.

Therefore, in this study, our objective was to evaluate the associations between gestational exposure to flood and pregnancy loss by matching a multi-region survey on maternal and child health in developing countries with spatial-temporal flood databases. Following this, we quantified the excess pregnancy loss attributable to gestational flood exposure in each selected country from 2010 to 2020.

## Results
### Characteristics of the matched cases
We utilized a matched case–control design to explore the relationship between flood events and pregnancy loss, which include miscarriages (<5 months of gestation) and stillbirths (>5 months of gestation). We collected all the potential individual-level data of women with records of pregnancy loss from multiple Demographic and Health Surveys (DHS) datasets worldwide[26]. Then, we matched the flood database with the spatial information of each individual's living cluster, which represented the geographic location of the village or neighborhood that the surveyed women lived in (see Methods for detail). By contrasting flood exposure conditions across various pregnancy outcomes for the same woman – spanning pregnancy losses to successful births—we assessed the potential impact of flood exposure on pregnancy outcomes.

Following the data selection process (Fig S7), we identified a total of 35,181 eligible cases from 33 developing countries out of 43 surveyed countries. This included women who had experienced both fruitful pregnancies and pregnancy losses and had been exposed to floods during past pregnancies. Their geographic distribution is illustrated in Fig.1. Among these women, 69,480 pregnancy losses (i.e., cases) were matched to 194,409 successful deliveries (i.e., controls) (Table 1). On average, each case of pregnancy loss was matched with 4.52 controls, and the average wait time between cases and controls was 24.1 months. Geographically, South Asia (43.4%) has the most cases, followed by Sub-Saharan Africa (27.8%) and South America (12.6%). Pregnancy loss occurred more frequently in rural (59.0%) than in urban areas (41.0%). Other descriptive statistics on the country- or survey-specific cases are summarized in Table S2. Flood exposure averaged 25.6 days, with a standard deviation of 25.3. As shown in Table S3, a total of 224 flood events were matched with the records of pregnancy loss, the majority were pinpointed to South Asia (25.0%) and Sub-Saharan Africa (38.8%). Furthermore, these regions also reported elongated flood durations, particularly Sub-Saharan Africa (59 days), South Asia (38 days), and South America (37 days).

### Association between flood exposure and pregnancy loss
To associate maternal flood exposure with pregnancy loss, we employed a conditional logistic model[27,28] (see Methods for detail). Broadly, we found positive and significant associations of gestational flood exposure with total pregnancy loss and two specific subtypes. As illustrated in Fig. 2, the odds ratio (OR) of pregnancy loss associated with gestational flood exposure was 1.08 (95% confidence interval (CI):1.04–1.11) with comparable odds for miscarriage (OR: 1.05, 95% CI:1.00–1.10) and

**Table 1 | Summary statistics of surveyed women who have reported at least one successful pregnancy before or after the pregnancy loss, and have experienced flood exposure one or more times during their previous gestation periods**

| | Total (n = 35,181) | Pregnancy loss (n = 69,480) | Successful delivery (n = 194,409) |
|---|---|---|---|
| **Region** | | | |
| Central American and Caribbean | 3335 | 6592 | 17,468 |
| South America | 4206 | 8747 | 21,990 |
| North Africa | 235 | 679 | 2642 |
| Sub-Saharan Africa | 9212 | 19,282 | 67,670 |
| Southern Africa | 864 | 2615 | 9863 |
| South Asia | 16,653 | 30,130 | 69,984 |
| West Asia | 676 | 1435 | 4792 |
| **Wealth** | | | |
| Poorest | 6991 | 13,489 | 46,503 |
| Poorer | 7501 | 14,809 | 45,255 |
| Middle | 7280 | 14,689 | 40,201 |
| Richer | 6874 | 13,711 | 34,379 |
| Richest | 6535 | 12,782 | 28,071 |
| **Education** | | | |
| Primary or on education | 18,332 | 35,235 | 124,125 |
| Secondary | 13,273 | 26,789 | 56,858 |
| Higher | 3576 | 7456 | 13,426 |
| **Location** | | | |
| Rural | 21,003 | 40,978 | 123,666 |
| Urban | 14,178 | 28,502 | 70,743 |
| **Water Source** | | | |
| Surface water | 14,363 | 26,631 | 67,308 |
| Intermediate | 14,170 | 28,838 | 86,714 |
| Piped/Tap | 3230 | 6458 | 22,333 |
| Other | 3246 | 6949 | 16,423 |
| Unknown | 9 | 13 | 22 |
| **Floor Material** | | | |
| Natural | 13,141 | 26,816 | 86,271 |
| Rudimentary | 2,759 | 4620 | 11,514 |
| Finished | 19,199 | 37,850 | 95,790 |
| Unknown | 75 | 167 | 738 |
| Mother's age at birth | 24 (20, 29) | 26 (22,32) | 23 (20, 27) |
| **Other environmental condition[a]** | | | |
| Average temperature during pregnancy (°C) | 25.74 (22.04, 28.47) | 25.38 (21.55, 28.24) | 25.85 (22.21, 28.55) |
| Average cumulative precipitation during pregnancy (units: m)[b] | 114.70 (20.23, 396.89) | 93.16 (18.19, 369.68) | 122.80 (21.04, 405.82) |

[a]Data are listed as median [IQR], IQR: interquartile range.
[b]From the first month of pregnancy to the matched pregnancy loss month.

stillbirth (OR: 1.08, 95% CI: 1.01–1.15). There was a significant association (OR: 1.12, 95% CI: 1.04-1.21) of flood exposure with early miscarriage (≤2 gestational months), but not with fetal miscarriage (>2 and ≤ 5 gestational months) (OR: 1.03, 95% CI: 0.96–1.11).

In the supplementary analysis on pre-conception flood exposures, we also found a positive and significant association of pregnancy loss with flood exposure during the 3 months preceding the conception (OR: 1.05, 95% CI: 1.02–1.08), and with flood exposure from 3 to 6 months before pregnancy (OR: 1.03, 95% CI: 1.01–1.06), but not with flood exposure from 6 to 9 months before pregnancy (OR: 1.03, 95% CI: 0.85–1.21).

Stratified analyses highlighted several distinct patterns. the ORs of pregnancy loss were significant for women who experienced prolonged flood exposure (>16 days, OR: 2.00, 95% CI: 1.83–2.18), and those exposed to flooding on multiple occasions (≥2 times, OR: 1.78, 95% CI: 1.69–1.88). In contrast, brief flood exposures, less than 16 days, (OR: 1.02, 95% CI: 0.76–1.30) and singular flood events (OR: 1.02, 95% CI: 0.98–1.06) did not showcase statistically significant impact. Pregnancy loss risk was found to be pronounced for women either below 21 years or above 35 years (age <21, OR: 1.12, 95% CI:1.02–1.24; age>35, OR: 1.17, 95% CI:1.07–1.28). However, no statistically significant association was observed for women aged 21–25 years (OR: 1.11, 95% CI:0.98–1.26) and 25–35 years (OR:1.02, 95% CI: 0.95–1.09). The highest risks of pregnancy loss were during mid-pregnancy (OR: 1.07, 95% CI: 1.02–1.15) and late pregnancy (OR: 1.05, 95% CI: 1.01–1.15), while early pregnancy showed no statistically significant risk (OR: 1.00, 95% CI: 0.93–1.08). Figure 3 depicts varying risks contingent upon socio-economic factors and living conditions. Women with limited wealth and educational attainments consistently recorded higher ORs. For those with the lowest income and education, the ORs were 1.12 (95% CI:1.01–1.23) and 1.11 (95% CI:1.04–1.20), respectively. In terms of living conditions, women who had access to surface water showed elevated ORs compared to those accessing intermediate or piped/tap water. Similarly, households with rudimentary flooring demonstrated a higher OR than those furnished with natural or finished floors.

Sensitivity analyses presented consistent findings. First, we derived similar ORs to our main estimates using the alternative flood database, although the number of matched cases reduced (Table S1 and Fig. S1). Second, the OR estimates did not vary considerably for models of different sets of covariates, except for the adjustment of maternal age (Fig. S2). Third, for different regions, our results are stable for each of the regions and are not relying on one single region (Fig. S3). Finally, by focusing on mothers residing in a consistent household for a decade or more, the ORs still remained significant.

### Pregnancy loss attributed to gestational flood exposure

We thereafter used the estimated ORs to calculate the excess numbers of pregnancy loss per 10,000 deliveries for each country involved in the study (Fig. 4). Throughout the 2010s (2010–2020), maternal exposure to gestational flood events may be responsible for -107,888 (95%CIs: 53,944–148,345) total excess pregnancy losses per year or 16 (95% CIs: 8–22) per 10,000 pregnant women per year across the selected 33 low and middle-income countries. South Asia had the highest proportion of excess pregnancy losses, particularly in its northern and eastern regions. Proportionally, Fig. 4 shows that countries from Central America and Caribbean, and South America suffered the largest burden of pregnancy loss due to flood exposure during the 2010s. Regarding specific flood types (Fig. S5), heavy rains or monsoon rains contributed to the largest proportion (89.84%) of flood-related pregnancy losses, followed by tropical cyclones (9.47%) and levee/dam failure (0.70%). Notably, an upward trend in the annual proportion of excess pregnancy losses was observed from 2010 to 2020 (Fig. 5).

### Discussion

More frequent and intense flood events have taken place worldwide over the past several decades due to anthropogenic climate change[5]. While a limited number of studies have delved into the effects of flood exposure on women's and children's health. We found that gestational flood exposure could significantly increase the risk of pregnancy loss (including miscarriage and stillbirth) among women from 33 low and middle-income countries. The ORs were larger for women outside the

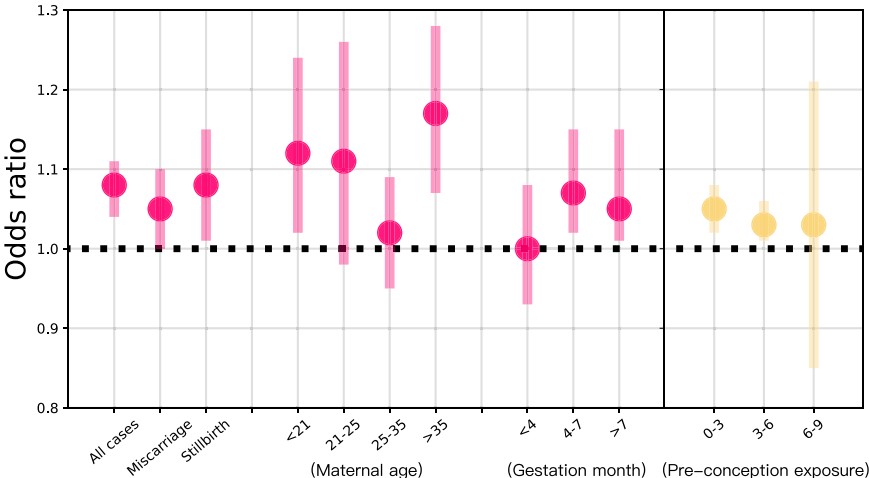

**Fig. 2 | The odds ratios of pregnancy loss for women associated with gestational flood exposure and pre-conception flood exposure.** Points are the estimated odds ratios of pregnancy loss (gestational flood exposure vs. non-gestational flood exposure), and error bars indicate 95% confidence intervals. A total of 35,181 eligible cases from 33 developing countries were included.

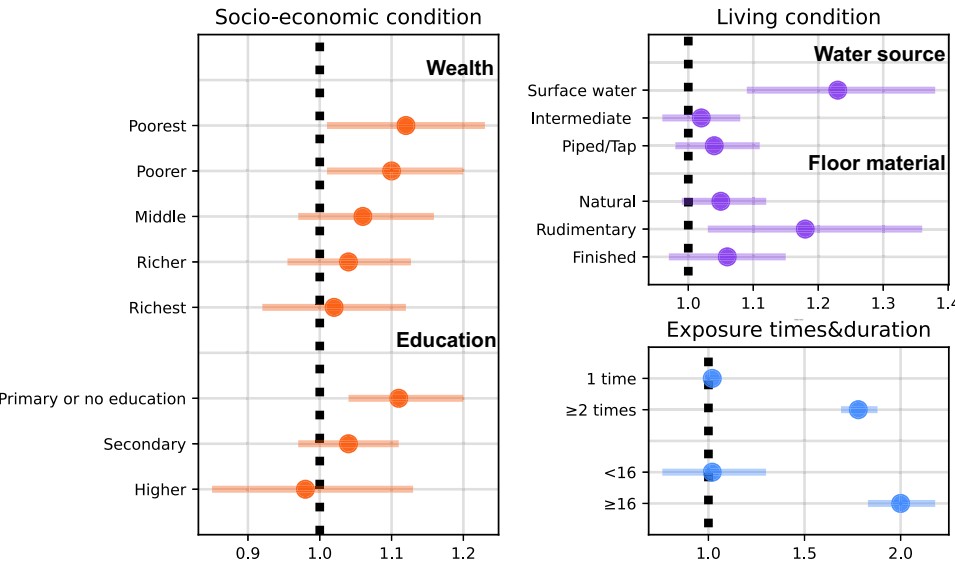

**Fig. 3 | Odds ratios of pregnancy loss associated with gestational flood exposure, classified by socioeconomic factors, living conditions, and flood duration.** Points are the estimated odds ratios of pregnancy loss (gestational flood exposure vs. non-gestational flood exposure) for each group, and error bars indicate 95% confidence intervals. Detailed case numbers for each group are provided in Table 1.

peak reproductive age range (<21 or >35 years) or those in their second trimester. Also, there were notable disparities in estimated risks based on socioeconomic factors and living conditions. We also observed a significant rise in pregnancy loss risk for women exposed to floods in the 0–3 months and 3–6 months prior to conception. Our estimations suggest that gestational flood events might account for -107,888 (CIs: 53,944–148,345) excess pregnancy losses per year across the selected 33 developing countries over the 2010s. This toll saw an escalation throughout the decade, with regions like Central America, the Caribbean, South America, and South Asia being predominantly affected. Maternal and child health is of utmost importance on the global agenda, evident from its inclusion in the United Nations Millennium Development Goals. Among the various pressing public health concerns, pregnancy loss emerges as a significant health concern, affecting families with limited resources in certain nations[29,30]. Hence, the epidemiological evidence we present in this study regarding the impact of gestational flood exposure on pregnancy loss assumes

critical significance in the context of global health, especially considering the backdrop of climate change.

Pregnant women are widely regarded as vulnerable to extreme weather events. While no prior epidemiological studies had directly linked flood events to pregnancy loss, our findings suggest a significant correlation between gestational flood exposure and increased risk of pregnancy loss for several reasons. Firstly, flood events could immediately induce or exacerbate accidental injuries, physiological stress, and the transmission of infectious diseases, which may directly cause pregnancy loss[24]. Secondly, essential services for pregnant women, such as facilities for safe labor and delivery, might be unavailable in areas affected by flood[14], and the unsafe delivery conditions may directly induce higher pregnancy loss risk, especially for pregnant women in the third trimester. Thirdly, flood-related destruction of the family and social structures, including households and local communities, leaves women more exposed to risks and to manage the risks alone. Fourthly, floods also lead to a range of indirect consequences.

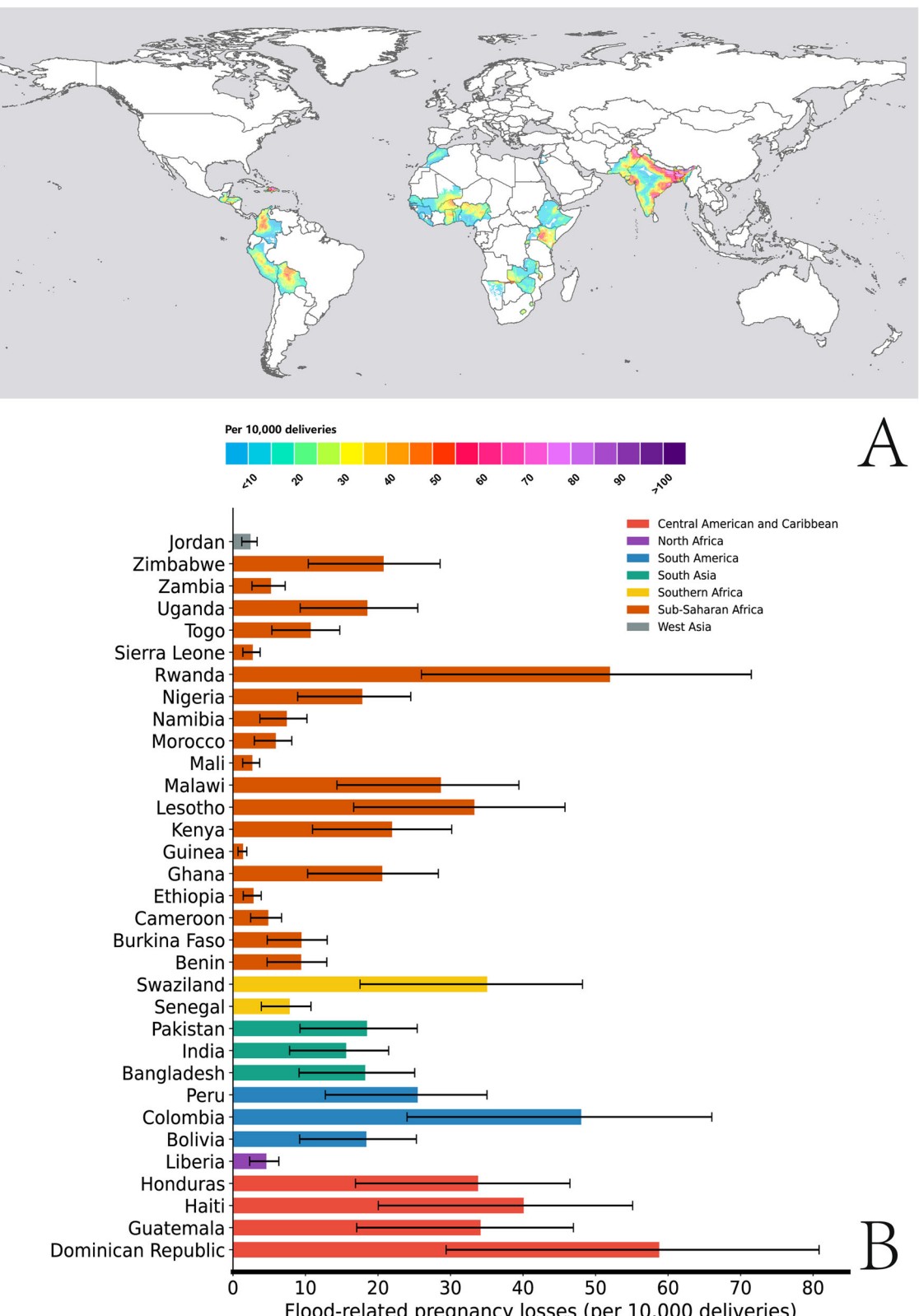

**Fig. 4 | Average annual excess pregnancy losses (per 10,000 deliveries) associated with gestational flood exposure in 33 developing countries (2010–2020). A** Spatial distribution of annual excess pregnancy losses (per 10,000 deliveries) at a spatial resolution of 10 km × 10 km. **B** Annual excess pregnancy losses (per 10,000 deliveries) by country. The bars represent the estimated flood-related pregnancy losses per 10,000 deliveries, error bars indicate 95% confidence intervals.

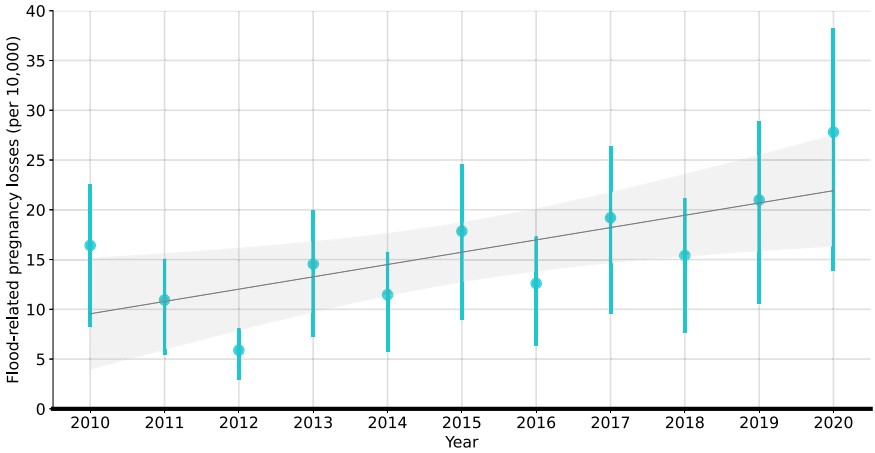

**Fig. 5 | Temporal trends in estimated annual excess pregnancy losses (per 10,000 deliveries) associated with gestational flood exposure in 33 developing countries from 2010 to 2020.** The points depict the estimated median values of annual excess pregnancy losses, with error bars representing 95% confidence intervals. The gray line illustrates an increasing trend in flood-related pregnancy losses as indicated by a simple linear regression model, with the shaded gray area representing the 95% confidence intervals of the simple linear regression results, based on the annual assessments.

For instance, severe water scarcity, more intense competition for water, and a more widespread epidemic of water-borne diseases[15]; flooding and polluted water resources may further lead to soil contamination, which can damage crops and disrupt food supplies[16]; Finally, pregnancy outcomes could be impacted by mental stress resulting from natural disasters during pregnancy[31], including flooding[21,22].

Our stratified analyses further elucidate the underlying mechanisms and the variability in population vulnerability concerning pregnancy loss due to maternal flood exposure. First, we found a significant increase in the risk of pregnancy loss among women residing in houses with rudimentary floor conditions, which refers to floors that are poorly constructed and lack proper sealing. Compared to houses with natural or finished floors, rudimentary floors are more prone to structural compromise during floods, rendering them less resilient to flood-induced challenges[32]. Consequently, this increased vulnerability can exacerbate the risks faced by pregnant women, including physical injuries and exposure to hazardous materials. Second, the increased risk observed among women with lower income and less education can be attributed to the barriers they might face in accessing adequate prenatal care, compared to women who are well-educated or financially privileged. Such barriers can profoundly impact how maternal health is managed during pregnancy, which could lead to negative outcomes like miscarriage or stillbirth[33]. Third, the increased risk among women with access to unsafe water sources suggests that floods can affect maternal health through water contamination. Floods contaminate water sources[34], disrupt sanitation infrastructure, and hinder drainage, creating an environment conducive to the spread of waterborne diseases and infections. Contaminated water and inadequate sanitation facilities increase the vulnerability to maternal infections, which are known risk factors for stillbirth or miscarriage[35]. Then, a pronounced rise in pregnancy loss risk was found in women younger than 21 or older than 35, with no significant influence observed for the age bracket between 21 and 35. For women younger than 21, some studies indicate a heightened risk of pregnancy complications due to the less physiological maturity or healthcare accessibility[36,37]. These factors make them more vulnerable to external stressors like floods. For women older than 35, advanced maternal age correlates with reduced fertility and a higher likelihood of chromosomal abnormalities in embryos, potentially resulting in miscarriage or stillbirth. Moreover, older women are more likely to have pre-existing health disorders, which may be exacerbated by flood-related stressors[36]. Furthermore, the significant risk seen in women during the mid to late

stages of pregnancy, compared to the early stage, might be due to the growing challenges related to mobility and relocating. In addition, the significant risks associated with longer durations and multiple times of flood exposure reflect heavier maternal physiological and psychological stresses originating from flood-related adverse factors, such as displacement, loss of resources, and disruption of healthcare services. Finally, the significant impact of pre-conception flood exposure underscores the long-term and sustained indirect impacts of floods for expectant mothers.

This multi-country study has several advantages. Firstly, our research provides new insights into the detrimental effects of floods on pregnancy outcomes, a frequently overlooked risk to women's and children's health, particularly in developing nations. Second, the ample findings from stratified analyses provide important clues about the vulnerability characteristics of pregnant women to flood exposure. This knowledge is invaluable in crafting specific protective measures for at-risk expectant mothers. Third, we present a country-specific estimation of the excess pregnancy losses in the past decade, aiding in the development of flood-related risk mitigation strategies and fortifying resilience to climate change in susceptible regions. Furthermore, the varied impact of different flood types across regions indicates the need for localized preventive strategies. Fourth, more frequent floods in low-latitude regions increase the burden of pregnancy loss in low-income and middle-income countries, highlighting the inequality for maternal and child health affected by climate-related disasters.

Some limitations of the present study should be also noted. First, the spatial data of the DHS clusters was intentionally altered to a degree to safeguard privacy, meaning that the precise location coordinates were scrambled. This could introduce inaccuracies when determining flood exposure. Second, potential residual confounding may arise from unaccounted maternal factors that evolve over time, such as chronic disease prevalence, medication usage, and shifts in income, healthcare, and housing conditions. Third, when estimating the excess pregnancy losses due to gestational flood exposure, we used the uniform ORs across different countries, as the limited sample size did not support a country-specific OR estimation. Fourth, we presumed the annual count of pregnant women in each grid remained constant from 2010 to 2020, given our access to only a single year of gridded data from WorldPop.

In conclusion, our multicounty, matched, case-control analysis offers strong and compelling evidence that gestational flood exposure significantly elevates the pregnancy loss risk for women from

developing countries. Furthermore, this risk is accentuated for women with lower income and educational levels, those relying on surface water, residing in urban settings, or living on a rudimentary floor. Our findings also highlight an escalating trend in the burden of pregnancy loss due to gestational flood exposure throughout the last decade, especially pronounced in LMICs in Central America and the Caribbean, South America, and South Asia. These insights should prompt the global community to prioritize and implement effective interventions, mitigating the adverse impacts of extreme weather events on maternal and child health, particularly in the face of a shifting climate.

## Methods

### Study design

This analysis consists of several steps. We first used a matched case-control design to estimate the relationship between flood events and pregnancy loss. In this process, we used individual-level data of women with records of pregnancy loss from multiple DHS datasets worldwide[26]. Thereafter, we presented a series of stratified analyses by different socioeconomic and living conditions. Finally, we used the estimated risk to project the numbers of excess pregnancy losses related to flood exposure for each country over the last decade (i.e., the 2010s).

### Pregnancy loss data

In this analysis, we collected all available DHS data on pregnancy loss. The DHS are nationally representative surveys that were routinely conducted at 3–5-year intervals in more than 90 developing countries[26,38]. Specifically, health outcomes including pregnancy loss and its incidence time were recorded in the reproductive calendar, which contains the monthly birth information for each of the respondents. Furthermore, these records were validated by a separate questionnaire (including questions about whether a pregnancy had been terminated, when it happened, etc.). Based on these validated records, we can extract some key pregnancy variables including the date of pregnancy outcomes (pregnancy loss or live birth) and the corresponding gestational periods. We extracted these individual data from the surveys relating to women who were reported to have suffered one or more times of pregnancy losses (defined as a case) and have one or more children alive at the time of screening (defined as a control). Pregnancy loss was defined as a miscarriage (<5 months of gestation) or stillbirth (>5 months of gestation)[27,28]. We also obtained other demographic and socioeconomic information. Latitude and longitude information for the clusters of the participants were extracted from the database, which represented the spatial location of the community that the surveyed women lived in. The trained field interviewers used the global positioning system devices to identify the central point of each cluster's populated area. The coordinates of each cluster are mostly displaced by up to 2 km in urban areas, and 5 km in rural areas to protect privacy[39]. For other individuals- and household-level information, we also collected other important variables on socioeconomic and living conditions, including maternal age, wealth, education, type of residential area (urban or rural), water source, and floor material of the living building. There is a total of 130 separate surveys with geographic information covering 56 LIMICs countries available for this analysis. Exclusion criteria are: (1) without spatial and temporal variables (reproductive calendar date and cluster location); (2) without delivery information or abnormal delivery record (Supplementary Material). According to the exclusion criteria, there remain 80 surveys in 43 developing countries between Jan 1, 1992, and March 31, 2020, which recorded a total of 90,465 cases of pregnancy loss from 44,847 clusters. In addition, based on the spatial information of each cluster, we also extracted and calculated the gestational length-adjusted mean temperature and precipitation from the ERA5 dataset, which is a widely used gridded reanalysis dataset produced by the

European Centre for Medium-Range Weather Forecasts with a 0.25 × 0.25-degree spatial resolution[40].

### Flood exposure data

We obtained information on flood events from the Dartmouth Flood Observatory (DFO) database, which has been widely used in flood exposure and impact assessment studies[41,42]. This database includes all verified flood events that have occurred since 1985, as reported by local news, governmental sources, or FloodList (http://floodlist.com/). The database contains dates and affected areas of the 4712 flood events that have occurred worldwide since 1985. A key feature of this dataset is that it includes the main causes of floods, which were simplified into four categories: heavy rainfall, tropical cyclones, levee/dam failure, and snow melt (it was excluded from the subtype analysis due to a very small number of events).

To allow for the sensitivity analysis, we used the Global Flood Database (GFD) dataset as an alternative source of flood exposure[1]. For this database, daily satellite imageries at a 250-m resolution were adopted to further depict the areas with inundation from all the flood events recorded in DFO dataset at a fine scale; in total, it estimated 913 flood events and their spatiotemporal extends from 2000 to 2018 worldwide. We did not utilize this database for the primary analysis, as satellite imagery was unable to capture certain flood events due to persistent cloud cover. Furthermore, it should be noted that the GFD dataset does not provide comprehensive coverage of all areas affected by flood events. Instead, it focuses on specific regions with flood inundation[1].

We matched the flood database with the spatial information of each individual's living cluster by adopting spatial analysis tools and statistical tools from the *geopandas* (version 0.11.0) and *pandas* (version 1.4.1) packages from the Python platform (version 3.8.10).

### Statistical analyses

We used a matched case–control design to explore the association between flood exposure and pregnancy loss following the methodology that was reported previously[27,28]. Using the records of mothers who had experienced pregnancy loss and successful delivery, we compared multiple pregnancy outcomes with each other from the same woman. First, for a case, we defined the gestational flood exposure (binary variable) when the spatial location of the residing cluster during the pregnancy period overlapped with the affected area by a flood event during the flooding period. For the corresponding controls, we calculated the maternal flood exposure (binary variable) according to the gestation length of the matched case. We provide an example to better illustrate our case-control design. As depicted in Fig. S6, a woman has three pregnancy records, in which the first and third pregnancies resulted in full-term vaginal deliveries, and the second pregnancy ended in a miscarriage in the sixth month of gestation. To determine the flood exposure for each pregnancy cycle, we have adopted a 6-month time window, starting from the beginning of each pregnancy month. This approach ensures that we capture the relevant exposure period for each pregnancy record. Similar to case-crossover studies, this matched case-control design could control for all confounders that vary between individuals as they remain constant between pregnancies of the same mother.

We applied a conditional logistic model to associate maternal flood exposure with pregnancy loss[27,28]. The main model included pregnancy loss or its subtypes (miscarriage and stillbirth) as the dependent variable and flood exposure as the independent variable. We adjusted for maternal age in the delivery year, and two categorical terms for the year and the month of conception in the main model to control for possible long-term and seasonal trends in pregnancy loss. Moreover, considering that the risk of miscarriage may escalate following previous adverse pregnancy outcomes in a woman[43], we also adjusted for a continuous variable representing the number of

previous pregnancy losses in the main model. To control for the possible confounding effects of climatic factors, we used natural spline functions with three degrees of freedom for the gestational-length mean temperature and precipitation, respectively. Specifically, for all the cases and controls, we calculated the mean temperature and precipitation from the starting month of pregnancy to the matched gestational months, which were obtained from the ERA5 (the fifth generation ECMWF atmospheric reanalysis of the global climate) dataset. Other individual- or household-level factors (e.g., maternal education level or household wealth level) were not adjusted in this analysis, because the factors were unchanged between pregnancies. Finally, we calculated the odds ratio (OR) and 95% confidence interval (CI) of pregnancy loss associated with maternal flood exposure with the adjustment for maternal age, temperature, precipitation, and time trends.

As a supplementary analysis, we also explored the impact of pre-conception flood exposure, which was determined when the spatial location of the residing cluster overlapped with the affected area by a flood event before pregnancy. The pre-conception flood exposure was categorized into three specific time windows: from preceding 3 months to the start of pregnancy (i.e., 0–3 months), from preceding 6 months to 3 months (3–6 months), and from preceding 9 months to 6 months (6–9 months). To avoid the potential overlaps, we excluded cases in the category of the pre-conception period of 0–3 months where flood exposure persisted after the pregnancy began. Similarly, any flood events that extended into the 0–3 period were disregarded in the exposure category of 3–6 pre-conception. The same principle was applied to the exposure category of 6–9 pre-conception.

In addition, we conducted several stratified analyses. First, we examined the association between gestational flood exposures and pregnancy loss in subgroups of flood durations longer than the average and shorter than the average, and floods with single-time exposure and multiple times exposure. As women younger than 21 or older than 35 years are at increased risk for adverse pregnancy and birth outcomes, we examined the association in subgroups with different maternal ages, including <21, 21–25, 25–35, and >35 years. Third, we also examined the association among women from different pregnancy periods, including early pregnancy (<4 month), mid-pregnancy (4–7 month), and late pregnancy (>7 month). Fourth, we conducted subgroup analyses by socioeconomic conditions, including wealth (the poorest, poorer, middle, richer, and the richest[44]) and education levels (primary or no education, secondary, and higher than secondary). Lastly, we examined the associations in subgroups of different living conditions, including residential types (urban and rural), water source (surface water, intermediate, and piped/tap), and floor materials (natural, rudimentary, and finished).

Furthermore, we calculated the excess numbers of pregnancy losses specifically attributable to gestational flood exposure. Firstly, the number of pregnancies with a 1 km spatial resolution for each country was obtained from the WorldPop database[45,46], which was further aggregated to a 10 km grid. Secondly, we linked this pregnancy database to the DFO database. Thirdly, we estimated the excess pregnancy losses attributable to each flood event following the equation:

$$EP_{iy} = (RR - 1) \times F_{iy} \times P_{iy}$$

In this formula, $i$ represents a grid $i$, $y$ is the studied year, and $EP_{iy}$ is the excess pregnancy losses in grid $i$ over year $y$; RR is the relative risk of pregnancy loss associated with flood exposure, and we assumed that RRs could be well represented by ORs calculated in this analysis because of the low prevalence of pregnancy loss; $F_{iy}$ represents the identified total number of flood events in grid $i$ over the year $y$; $P_{iy}$ is the corresponding number of pregnancy losses in grid $i$ during the

year $y$, which is calculated from the total number of pregnancies in grid $i$, and the average ratio of pregnancy loss for the region that this grid $i$ belongs to[29]. Accordingly, we estimated the number of excess pregnancy losses attributable to gestational flood exposure for each year from 2010 to 2020. Finally, we depicted the number of pregnancy losses (per 10,000 pregnancies) for each grid from 2010 to 2020 and summed over each country and three subtypes of flood events (heavy rainfall, tropical cyclones, and levee/dam failure).

We performed several sensitivity analyses. First, we re-analyzed the association using the alternative data source of flood events (GFD) that focused on the areas with flood inundation. Second, we reduced the covariates one for a time to examine the robustness of each of the covariates. Third, we tested the sensitivity of our main results by excluding each region from the analysis. Finally, extreme floods may force migration, resulting in a disparity between the house of women's interviews and the birth records; thus, we limited our sample to mothers who reported residing in the same house for a minimum of 10 years prior to the survey.

We fit the conditional logistic models using the function *clogit* in the R platform (version 4.0.3). All tests were two-sided with an alpha error level at 0.05.

## Reporting summary

Further information on research design is available in the Nature Portfolio Reporting Summary linked to this article.

## Data availability

Survey data including pregnancy history records and socioeconomic data in this study are publicly available upon request from the Demographic and Health Surveys Program (https://dhsprogram.com/). Flood data are also publicly available from The Flood Observatory (https://floodobservatory.colorado.edu/). The number of pregnancies with a 1 km spatial resolution for each country was obtained from the WorldPop database (https://hub.worldpop.org/geodata/listing?id=19). Source data are provided with this paper.

## Code availability

R codes for statistical analysis are available from the corresponding authors, Haidong Kan (kanh@fudan.edu.cn). We will respond to the requests within 2 weeks.

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

## Acknowledgements

H.K. is supported by the National Natural Science Foundation of China (92043301), the Shanghai Municipal Science and Technology Commission (21TQ015), and the Shanghai International Science and Technology Partnership Project (No. 21230780200); C.H. is supported by the Alexander von Humboldt Foundation for the Humboldt Research Fellowship.

## Author contributions

C.H. and Y.Z. are joint first authors. H.K. and R.C. contributed equally to the correspondence work. C.H. and Y.Z. analyzed the data and drafted the manuscript. H.K. and R.C. designed this work and revised the manuscript. C.H., Y.Z., J.B., and A.S. contributed to data collection and revision. All authors contributed to the development of the manuscript and approved the final draft. H.K. and R.C. are study guarantors. The corresponding author attests that all listed authors meet authorship criteria and that no others meeting the criteria have been omitted.

## Competing interests

The authors declare no competing interests.
