## [Peer Review File · Nature Communications]

Reviewers' Comments:

Reviewer #1:

Remarks to the Author:

The authors investigated a very interesting question: the association between maternal flood exposure and the risk of pregnancy loss and quantified the association using a dataset collected from 33 developing countries. Relevant evidence is limited yet. The authors further explored vulnerable groups and estimate excess pregnancy losses attributable to flood exposure, which would be instructive evidence for preventive intervention, disaster response, and prioritizing resources. Following are some suggestions to improve the paper.

Major comments:

1. Study design: one important issue is that, in such case-only study designs (e.g., case-crossover design), in which each individual is compared with his/herself, one must assume that within an individual, repeated events are independent of each other. However, in the case of a history of miscarriages, a woman may undergo tests to identify any underlying causes and take necessary precautions to prevent future miscarriages. The risk of miscarriage shows a strong pattern of recurrence, and is also increased after some adverse pregnancy outcomes (BMJ 2019;364:l869). Therefore, repeated events should not be independent of each other, which may lead to a biased estimate of the association.

2. Exposure assessment: The authors defined "a binary indicator of gestational flood exposure according to whether the residential cluster location during the pregnancy period was within the affected areas of flood events". This is fine with the DFO database because the spatial information provided by this database represent the areas affected by flood events. However, the spatial information provided by the GFD represents areas with inundation, which is different from flood-affected areas. The author should consider this difference when defining the exposure using the GFD and when interpreting the results of the sensitivity analysis.

3. Exposure assessment: The GFD dataset only captured about 30% of the global flood events. In this occasion, the gestational exposures of some participants may be misclassified as unexposed. How did the author cope with this?

4. Exposure assessment: It is stated, "For the corresponding controls, we calculated the maternal flood exposure (binary variable) according to the gestation length of the matched case." The authors should make it clear how you define exposure windows for controls? For example, if the gestation length of the matched case is 5 months, exposure windows should start from the first day of the last menstrual period rather than ends by the date of pregnancy outcomes.

5. Exposure assessment: Is it possible for a woman to experience more than one flood event during her pregnancy? If yes, the authors should consider the multiple exposures.

6. Exposure assessment: The authors should consider the duration of each exposure. What is the average length of flood events? The impact of a flood lasting for one day versus a flood lasting for one month should differ.

7. Statistical analysis: the authors should further adjust pregnancy history in the main model, as the risk of miscarriage shows a strong pattern of recurrence, and is also increased after some adverse pregnancy outcomes (BMJ 2019;364:l869).

8. Statistical analysis: "Similar to case-crossover studies, the matched case-control design could control for all confounders that vary between individuals as they remain constant between pregnancies of the same mother". What is the wait time between case (pregnancy loss) and self-controls (successful delivery)? It's suggested to wait for at least 18 months between giving birth and getting pregnant again. Have the authors considered any changes for the mother during this period? For example, maternal age, chronic disease, medicine use. The authors should try to justify that individual- or household- level factors remained constant between conceptions.

9. Statistical analysis: For the methodology employed to evaluate the excess numbers of pregnancy losses, please note that the pregnancy losses attributable to gestational flood exposure does not fully represent the pregnancy losses attributable to flood events (e.g., pregnancy losses attributable to pre-pregnancy exposure).

10. Method: Please define "gestational length-adjusted mean temperature and precipitation". Unexceptionally, "gestational mean temperature and precipitation" rather than "gestational length-adjusted mean temperature and precipitation" were adjusted in the main model.

11. Method: authors should also consider pre-conception flood exposure. One of the most immediate impacts of flooding is damage to property and infrastructure, which can disrupt daily life and lead to financial losses. The consequences of flooding can also persist over the long-term, as the reconstruction of homes, businesses, and other structures can be a protracted process. Furthermore, floods can cause huge economic ramifications, for example, diminishing the output of crops and livestock, especially in developing countries. Finally, flood may have an impact on mental health. Therefore, a woman may experience flooding prior to becoming pregnant, yet still be impacted by the health and economic burdens caused by flooding. Since the aim of this study is to analyze the impact of maternal exposure, and excess pregnancy losses were estimated, the pre-conception exposure should be considered. The following study may be helpful.

Guo, C., Chen, G., He, P., Zhang, L., & Zheng, X. (2020). Risk of cognitive impairment in children after maternal exposure to the 1998 Yangtze River flood during pregnancy: analysis of data from China's second National Sample Survey on Disability. *The Lancet. Planetary health*, 4(11), e522–e529. [https://doi.org/10.1016/S2542-5196\(20\)30198-4](https://doi.org/10.1016/S2542-5196(20)30198-4)

12. Stratification analysis: Miscarriage (< 5 months of gestation) can be further classified as embryonic loss (or early miscarriage) when it occurs before 10 gestational weeks and fetal loss (or fetal miscarriage) when it occurs after 10 gestational weeks. Would floods have different effect on embryonic loss and fetal loss?

13. Results: the summary statistics of flood exposure is missing.

14. Results: the authors didn't cite Table 1 in text and the description of Table 1 is missing.

15. Results and discussion: I suggest not talking about the strength of the relationship between urban and rural, as based on the size of the effect ORs are similar. The authors ought to present the statistical tests and display the p-value indicating the difference in odds ratios between subgroups.

16. Discussion: what is the rationale behind the authors describing individuals under 25 years of age as having a non-optimal age for pregnancy?

17. Discussion: can the authors provide a more detailed explanation for the higher odds ratios observed among pregnant women below the age of 25 or above 29, particularly during the second trimester?

Minor comments:

1. Introduction: "The health effects of flood exposure have been well-documented for the general population, including injuries, communicable diseases, and vector-borne diseases." The health effects of flood exposure on the general population are actually not well-documented, where the outcomes are limited to psychological disorders, infectious disease, and gastrointestinal diseases.

2. Introduction: The first paragraph mainly focuses on the indirect health outcomes, and the indirect impacts on vulnerable groups. The author should establish the plausibility of the association between flood and pregnancy loss. Here is a study may be helpful: Guo, C., Chen, G., He, P., Zhang, L., & Zheng, X. (2020). Risk of cognitive impairment in children after maternal exposure to the 1998 Yangtze River flood during pregnancy: analysis of data from China's second National Sample Survey on Disability. *The Lancet. Planetary health*, 4(11), e522–e529. [https://doi.org/10.1016/S2542-5196\(20\)30198-4](https://doi.org/10.1016/S2542-5196(20)30198-4)

3. Introduction: "The existing epidemiological evidence for the relationship between maternal flood exposure and pregnancy loss is limited to small sample sizes, single flood events, and specific regions^{11,16}". The references are not relevant with pregnancy loss.

4. Introduction: "The existing epidemiological evidence for the relationship between maternal flood exposure and pregnancy loss is limited to small sample sizes, single flood events, and specific regions^{11,16}, making it difficult to draw reliable conclusions." This should not be the leading sentence of this paragraph, which focuses on research in LIMICs. The author may consider move this sentence to the last paragraph.

5. Introduction: "and is also helpful for prioritizing approaches to deliver crucial services". This is vague.

6. Method: "We also obtained demographic and socioeconomic information on the individual and household levels". Is this a leading sentence of the following information? If yes, the latitude and longitude information is on a village level rather than an individual or household level. If no, please specify the individual- and household-level information.

7. The authors mentioned 43 countries in the method section, whereas the title states 33 countries.

8. Method: "We did not utilize this database because satellite imagery could not detect some flood events due to persistent cloud cover". I guess the author would like to say "we did not utilize this database for the primary analysis".

9. Analysis: "we defined the gestational flood exposure (binary variable) when the spatial location of the residing cluster during the pregnancy period was overlapped with the affected area by a flood event during the flooding period". This has been defined in the section flood exposure data.

10. Analysis: Please define "pregnancy periods", which was used in the stratified analyses.

11. Table 1: the subheading "Floor material" should not include "the mother's age at birth".

12. The authors should rephrase "birth month" and "birth year" in Fig S2 in another way? The term "birth" should not be used where the outcome is pregnancy loss.

13. Fig S2: the title of Fig S2 "different sets of controls", should it be "different sets of covariates"?

14. Fig S3: what is the meaning of the dashed line?

15. Does the wealth presented in Figure 3 correspond to the income level in section "Pregnancy loss data"?

16. The floor material was categorized as natural, unfinished, and finished in Table 1, but as natural, rudimentary, and finished in text. Please ensure consistency of the terminology used.

Reviewer #2:

Remarks to the Author:

The authors performed a matched case-control study in an attempt to correlate an increased risk of pregnancy loss to flooding events during pregnancy in 33 developing countries. Individual-level data of women with pregnancy loss were retrieved from multiple DHS datasets collected between 2010 and 2020 including 69,480 pregnancy losses and 194,409 control with successful deliveries. Information about flooding events was primarily collected from the Dartmouth Flood Observatory Database.

An overall odds ratio (OR) of 1.06 (95% CI 1.02-1.10) for pregnancy loss in flood-exposed

pregnancies was found. Women with low income and poor education living in urban areas with an unfinished floor were in a particular high risk of pregnancy loss. The risk of pregnancy loss was only significantly increased between gestational month 4-7 and only women younger than 25 and older than 29 years had an increased risk of pregnancy loss.

The study is interesting and original, but the validity of the results is highly dependent on the quality of the DHS data and the possibility to correlate data from the Flooding Database with the actual flooding exposure during the women's pregnancies. I am not able to assess the quality of the data from the two databases.

The information about pregnancy losses that have happened between gestational month 4-7 must be considered valid since the registration of late pregnancy losses (in contrast to earlier losses) is probably complete also in developing countries.

There is some confusing information regarding the selection of cases in the results section: 35,181 cases were identified with at least one successful pregnancy before or after the pregnancy loss. Why should cases have given birth in addition to having experienced pregnancy loss? It is also unclear what is the 3.72 months interval between matched pregnancies in the case and control group. Is it the interval between date of pregnancy loss in cases versus date of delivery in the control group? I would like inclusion of a figure illustrating how cases and controls were identified and matched.

I would like a discussion about the results shown in fig 2 that only women with age < 25 or > 29 years are in increased risk of pregnancy loss when exposed to flooding. Could this finding be an artefact and be due to a methodological error? I would like a further subdivision according to age e.g. < 21, 21-25, 25-29, 29-33 and 33-37 years. A more detailed subdivision will clarify which age groups are in a particular increased risk of pregnancy loss.

Reviewer #3:

Remarks to the Author:

Review of "Flood exposure and pregnancy loss in 33 developing countries" (NCOMMS-23-04056-T)

This manuscript presents an analysis of over 90,000 records of pregnancy loss from over 30 developing countries that have experienced incidents of major flooding between 2010-2020. Results reported include a 1.06 OR for pregnancy loss associated with floods occurring at the time of gestation. This study provides a unique analysis of understudied populations comparing the outcomes of women's non-flood related pregnancies with those that occurred during floods. Furthermore, the relevance of the sample (developing country residents) and topics (flooding and pregnancy) are well established.

My enthusiasm is mitigated, however, by a lack of clarity, especially involving the methods, and clear links between suggested mechanisms and outcomes. The use of undefined technical terms in sections prior to the methods is problematic. The independent understandability of the results section should be improved. I provide specific examples of this in the itemized list below.

Another general issue involves the mechanisms highlighted by the authors, some are mentioned in the introduction, but a more comprehensive list begins on line 185. Using stratified analyses the authors claim to have identified possible "mechanisms and population vulnerability for the risks of pregnancy loss induced by maternal flood exposure." The paragraph goes on to list lower income, less education, less preparation for flood events, informal settlements, inefficient drainage, unplanned sanitation infrastructure, and unsafe water. However, there is no discussion of evidence relating these factors to stillbirth or miscarriage. Please explain how these factors might contribute to the specific outcomes analyzed in this report and provide evidence for these associations.

Related to the above issue, the authors also identify "unfinished floor" as a risk factor and as support for their hypothesis that "flood-related accidental injuries, harm, and physiological stress may be the dominant causes of pregnancy loss." Please explain how "unfinished floor" indicates that there is greater incidence of accidents, harm, physiological stress, etc. and how the data indicate that these are "dominant" causes. Also, please clarify the theoretical rationale for the hypothesis.

Finally, I was surprised that there was no mention of psychological stress as a possible mechanism. Research has shown that pregnancy outcomes are impacted by the stress resulting from natural disasters during pregnancy (e.g., Dancause et al., 2011), including flooding (Hilmert et al., 2016; Tong et al., 2011). Many of the potential mechanisms, particularly SES-related factors, are stress-related.

Specific Items:

ABSTRACT

Please clarify the following:

The significance of "unfinished floor" is not clear.

"some indirect floods impact pathways" is unclear

"nearly straight upward trend" is imprecise and unclear.

INTRODUCTION

Please explain what is meant by "may further decrease the resilience of pregnant women." Is this at an individual level, a social resource availability level, or other?

Please explain what is meant by "is important for global public health." Will understanding how floods in these specific areas help us better understand the impact of flooding globally? Or does this statement mean something else? I wonder if it would be more effective to focus on the importance of better understanding these understudied, at risk populations in order to help those populations in particular.

RESULTS

In general I found this section very difficult to follow when read prior to the methods. In general, technical terms need to be defined or more descriptive terms need to be used. For example,

Please define "pregnancy loss" in the results.

Please define "living cluster" or use a more descriptive term to improve clarity.

Please explain the research design used when describing the sample on lines 97-103. It is not clear what is being matched, whether it is between or within subjects, how 4.52 controls are matched if it is within-subject. Alternatively, the authors could consider moving this information to the Methods.

Lines 140-142, please provide the proportion of pregnancy losses estimated to be flood related relative total pregnancy losses for those areas.

Please provide a rationale for analyzing the impact of different flood types in lines 146-148. Are these differences simply due to base rate frequencies?

DISCUSSION

Please explain "randomly scrambled" (line 221)

METHODS

Please clarify the following,

"modification analyses." Is this the same as moderation?

"reproductive calendar"

"clusters of participants"

Please explain why the time period for this study was chosen when the methods says that data were available for Jan 1, 1992 to March 31, 2020 (Line 280)

The paragraph beginning on line 298 describes a database and then says "We did not utilize this database..." Please clarify the purpose of this paragraph.

The Methods section includes definitions that clarify some of the issues in the prior sections, however it is still not clear to me if there were only within-subjects "matched" comparisons or between-subjects comparisons as well. Some of the results suggest that there were between-subject analyses used, and if this is the case, please clarify how this was done.

SUPPLEMENTARY MATERIAL

The supplementary material had similar clarity issues to those mentioned previously. For example, stating that Figure S3 "confirmed that our results are stable..." would be clearer with an explanation. Also, terms like "per grid square," "estimated distributions of pregnancies," "spatial patterns of estimates," and the methods described in section 2 are unclear without more precise definitions and explanation.

Please note that the sentence concluding the paragraph in this section on line 77-79 seems to be out of order with the first sentence of the next paragraph. That is it seems like the authors "estimate the excess pregnancy losses related to flood exposure" before they "identified whether pregnancies... were exposed to flood events."

Reviewer #1 (Remarks to the Author):

The authors investigated a very interesting question: the association between maternal flood exposure and the risk of pregnancy loss and quantified the association using a dataset collected from 33 developing countries. Relevant evidence is limited yet. The authors further explored vulnerable groups and estimate excess pregnancy losses attributable to flood exposure, which would be instructive evidence for preventive intervention, disaster response, and prioritizing resources. Following are some suggestions to improve the paper.

Major comments:

1. Study design: one important issue is that, in such case-only study designs (e.g., case-crossover design), in which each individual is compared with his/herself, one must assume that within an individual, repeated events are independent of each other. However, in the case of a history of miscarriages, a woman may undergo tests to identify any underlying causes and take necessary precautions to prevent future miscarriages. The risk of miscarriage shows a strong pattern of recurrence and is also increased after some adverse pregnancy outcomes (BMJ 2019;364:l869). Therefore, repeated events should not be independent of each other, which may lead to a biased estimate of the association.

Response:

Thank you for this insightful comment regarding the issue of independence in case-only study designs. We acknowledge the potential influence of previous adverse pregnancy outcomes on the risk of miscarriage and agree that it can introduce bias into the association estimation.

To address this concern and consider the suggestion from the reviewer in comment NO.7, we have made the following modifications to our methodology. Firstly, we calculated the cumulative number of pregnancy losses for each record from each participant based on all previous records. This allowed us to capture the history of pregnancy losses experienced by each individual. Then, we added a variable representing the number of previous pregnancy losses as a covariate in the main model, which allowed us to account for the potential confounding effect of miscarriage following previous adverse pregnancy outcomes.

We found the association between gestational flood exposure and pregnancy loss remained significant after this adjustment. Specifically, the odds ratio for pregnancy loss associated with gestational flood exposure was 1.08 (95% confidence interval: 1.04-1.11).

Furthermore, we conducted a sensitivity analysis to test the impact of adjusting for pregnancy history on the main results. These tests were designed to assess whether this adjustment could significantly affect the associations we observed.

Please refer to lines 397 to 400 in the statistical analysis, lines 132 to 134 in the result section, and section 1.4 of the supplementary material.

2. Exposure assessment: The authors defined “a binary indicator of gestational flood exposure according to whether the residential cluster location during the pregnancy period was within the affected areas of flood events”. This is fine with the DFO database because the spatial information provided by this database represents the areas affected by flood events. However, the spatial information provided by the GFD represents areas with inundation, which is different from flood-affected areas. The author should consider this difference when defining the exposure using the GFD and when interpreting the results of the sensitivity analysis.

Response:

Thanks for this valuable comment regarding the differences in exposure assessment in the sensitivity analysis. We have carefully considered the difference between the GFD and DFO datasets and have made the necessary clarifications in the revised manuscript.

Firstly, in the Methods section, we have added clarification regarding the GFD dataset, as follows:

“For the GFD, daily satellite imageries at a 250-meter resolution were adopted to further depict the areas with inundation from all the flood events recorded in the DFO dataset at a fine scale.” (Lines 359 to 361)

“We did not utilize this database for the primary analysis, as satellite imagery was unable to capture certain flood events due to persistent cloud cover. Furthermore, it should be noted that the GFD dataset does not provide comprehensive coverage of all areas affected by flood events. Instead, it focuses on specific regions with flood inundation.” (Lines 363 to 367)

Then, we interpreted the results of this sensitivity analysis in the supplementary material (Section 1.1). We explicitly explained that we used the GFD data to evaluate the impact of exposure to flood inundation during the pregnancy period, not the impact of flood-affected areas. We thereby acknowledge the difference between the GFD and DFO datasets and added an explanation of the use of GFD data for sensitivity analysis. Please see the 1.1 section of the supplementary material.

3. Exposure assessment: The GFD dataset only captured about 30% of the global flood events. In this occasion, the gestational exposures of some participants may be misclassified as unexposed. How did the author cope with this?

Response:

To address the concern regarding the potential misclassification of gestational exposures in our sensitivity analysis, we have implemented additional screening measures during the matching process.

Firstly, we excluded any records of women that overlapped with a flood event in the Department of Flood Observations (DFO) but not in the Global Flood Database (GFD) from this analysis. By doing so, we ensured that the records included in this part of the sensitivity analysis fell into one of the following categories: records with no exposure to floods, records exposed only to floods identified by the GFD, or records demonstrating the simultaneous recognition of flood events from both the GFD and DFO. Our aim in implementing this procedure was to minimize misclassification that could arise from solely considering unexposed data based on the GFD dataset.

An additional explanation about this part of the methods has been added, please see section 1.1 in the supplementary material.

4. Exposure assessment: It is stated, “For the corresponding controls, we calculated the maternal flood exposure (binary variable) according to the gestation length of the matched case.” The authors should make it clear how you define exposure windows for controls? For example, if the gestation length of the matched case is 5 months, exposure windows should start from the first day of the last menstrual period rather than ends by the date of pregnancy outcomes.

Response:

We appreciate the suggestion for clarification. To address this concern more explicitly, we have included a detailed example in Fig S6 to illustrate the exposure assessment approach. In this example, a woman has three pregnancy records, in which the first and third pregnancies resulted in full-term vaginal deliveries, and the second pregnancy ended in a miscarriage in the sixth month of gestation. To determine the flood exposure for each pregnancy cycle, we have adopted a 6-month time window, starting from the beginning of each pregnancy month. This approach ensures that we capture the relevant exposure period for each pregnancy record. In other words, we used the month of the first day of the last menstrual period as the starting time for each pregnancy.

A related explanation about the concern has been added in the method part (lines 382 to 388).

5. Exposure assessment: Is it possible for a woman to experience more than one flood event during her pregnancy? If yes, the authors should consider the multiple exposures.

Response:

Yes, it is possible for a woman to experience more than one flood event during her pregnancy. According to the suggestion, we have added a stratified analysis to examine the association between pregnancy loss and floods of different exposure times.

The results from this additional analysis indicate that there is a significant association for floods with multiple times exposure (≥ 2 times, OR: 1.78, 95% CI: 1.69-1.88), but an insignificant association for floods with single-time exposure (OR: 1.02, 95% CI: 0.98-1.06).

Please refer to lines 412 to 422 in the methods section for a detailed description of the methodology, lines 137 to 142 for the specific results related to the different durations of floods, and lines 259 to 263 for the related discussion.

6. Exposure assessment: The authors should consider the duration of each exposure. What is the average length of flood events? The impact of a flood lasting for one day versus a flood lasting for one month should differ.

Response:

The average length of flood exposure is 25.6 days, with a standard deviation of 25.3 and a mean value of 16.0 days. This information has been added (lines 121 to 122)

According to the suggestion, we have added a stratified analysis to examine the association between floods of different durations and pregnancy loss.

The results from this additional analysis indicate that there is a significant association for floods with durations longer than the mean value (16 days) (OR: 2.00, 95% CI: 1.83-2.18), but an insignificant association for floods shorter than the mean value (OR: 1.02, 95% CI: 0.76-1.30).

Longer exposure to flood-related stressors, such as displacement, loss of resources, and disruption of healthcare services, may lead to heavier maternal physiological and psychological stress. Prolonged stress has been associated with adverse pregnancy outcomes, including pregnancy loss. So, it is plausible that the impact of these stressors over a longer period could contribute to a higher risk of pregnancy loss among women exposed to floods for longer durations.

Please refer to lines 423 to 425 in the methods section for a detailed description of the methodology, lines 143-145 for the specific results related to the different durations of floods, lines 259 to 263 for the related discussion.

7. Statistical analysis: the authors should further adjust pregnancy history in the main model, as the risk of miscarriage shows a strong pattern of recurrence, and is also increased after some adverse pregnancy outcomes (BMJ 2019;364:l869).

Response:

Done as suggested. We calculated the cumulative number of pregnancy losses for each participant based on their previous records. Then, we added a continuous variable representing the number of previous pregnancy losses as a covariate in our main model, which allowed us to account for the potential confounding effect of previous miscarriage. Please refer to lines 397 to 400.

Please also see the explanations in comment NO.1 in detail.

8. Statistical analysis: “Similar to case-crossover studies, the matched case-control design could control for all confounders that vary between individuals as they remain constant between pregnancies of the same mother”. What is the wait time between case (pregnancy loss) and self-controls (successful delivery)? It's suggested to wait for at least 18 months between giving birth and getting pregnant again. Have the authors considered any changes for the mother during this period? For example, maternal age, chronic disease, medicine use. The authors should try to justify that individual- or household- level factors remained constant between conceptions.

Response:

The average wait time between cases and controls is 24.1 months (SD: 4.8 months), We have now added this information in line 118.

To account for potential changes in time-varying individual-level factors, we had already considered the maternal age in the delivery year in the main model. Additionally, we had already included two categorical terms for the year and the month of conception in the main model to control for possible long-term and seasonal trends in pregnancy loss for the women. These methods have been widely used in previous studies to control for time-related confounders (1,2).

However, we acknowledge that there are some other uncontrolled individual-level factors due to the lack of data, such as chronic diseases and medicine use as mentioned by the reviewer. We have added this as one of the limitations of our research (lines 284 to 285).

Selected references:

1. Xue, T., Zhu, T., Geng, G. & Zhang, Q. Association between pregnancy loss and ambient PM_{2.5} using survey data in Africa: a longitudinal case-control study, 1998–2016. *The Lancet Planetary Health* 3, e219-ee225 (2019).
2. Xue, T., Tong, M., Li, J., Wang, R., Guan, T., Li, J., Li, P., Liu, H., Lu, H., Li, Y. and Zhu, T., 2022. Estimation of stillbirths attributable to ambient fine particles in 137 countries. *Nature Communications*, 13(1), p.6950.

9. Statistical analysis: For the methodology employed to evaluate the excess numbers of pregnancy losses, please note that the pregnancy losses attributable to gestational flood exposure does not fully represent the pregnancy losses attributable to flood events (e.g., pregnancy losses attributable to pre-pregnancy exposure).

Response:

Thanks for the suggestion. We agree that it is important to note that our estimation primarily focuses on the pregnancy losses attributable to gestational flood exposure, rather than encompassing all pregnancy losses associated with flood events, including pre-pregnancy exposure.

To address this concern and provide a more accurate explanation of our evaluation results, we have made substantial revisions throughout the manuscript. Firstly, we have added an explanation in the methodology to explicitly clarify the scope of our estimation, highlighting that our investigation specifically examines the impact of gestational flood exposure on pregnancy losses, and not flood exposure in general (lines 438 to 439). Then, we revised the relevant statements throughout the manuscript to emphasize that our estimation specifically represents the pregnancy losses directly attributable to gestational flood exposure.

10. Method: Please define “gestational length-adjusted mean temperature and precipitation”. Unexceptionally, “gestational mean temperature and precipitation” rather than “gestational length-adjusted mean temperature and precipitation” were adjusted in the main model.

Response:

To address this concern, we have provided a more concise and clear explanation of how we adjusted for gestational length in the main model.

Specifically, for this step, we aimed to adjust the temperature and precipitation data in the main model based on the starting month of pregnancy and the corresponding gestation months determined by the matched cases. We calculated the mean temperature and precipitation during this specific period according to the ERA5 dataset.

We have incorporated this explanation into the methodology section (lines 403 to 406).

11. Method: authors should also consider pre-conception flood exposure. One of the most immediate impacts of flooding is damage to property and infrastructure, which can disrupt daily life and lead to financial losses. The consequences of flooding can also persist over the long-term, as the reconstruction of homes, businesses, and other structures can be a protracted process. Furthermore, floods can cause huge economic ramifications, for example, diminishing the output of crops and livestock, especially in developing countries. Finally, flood may have an impact on mental health. Therefore, a woman may experience flooding prior to becoming pregnant, yet still be impacted by the health and economic burdens caused by flooding. Since the aim of this study is to analyze the impact of maternal exposure, and excess pregnancy losses were estimated, the pre-conception exposure should be considered. The following study may be helpful.

Guo, C., Chen, G., He, P., Zhang, L., & Zheng, X. (2020). Risk of cognitive impairment in children after maternal exposure to the 1998 Yangtze River flood during pregnancy: analysis of data from China's second National Sample Survey on Disability. *The Lancet. Planetary health*, 4(11), e522–e529. [https://doi.org/10.1016/S2542-5196\(20\)30198-4](https://doi.org/10.1016/S2542-5196(20)30198-4)

Response:

According to the suggestion, we have incorporated an analysis of pre-conception flood exposure as supplementary analysis, as the present study focused on gestational flood exposure. Specifically, we defined pre-conception flood exposure as situations where the spatial location of the residing cluster overlaps with an area affected by a flood event before pregnancy. We have categorized three specific time windows for pre-conception flood exposure: 0-3 months before pregnancy, 3-6 months before pregnancy, and 6-9 months before pregnancy.

To avoid the potential overlaps, we excluded cases where flood exposure persisted after the pregnancy began in the 0-3 pre-conception flood exposure category. Similarly, any flood events that extended into the 0-3 period were disregarded in the 3-6 pre-conception flood exposure category. The same principle was applied to the 6-9 pre-conception flood exposure category.

Results suggested the positive and significant association of the total pregnancy loss with flood exposure during the previous 3 months before pregnancy (OR: 1.05, 95% CI: 1.02-1.08), and during the previous 3-6 months (OR:1.03, 95% CI: 1.01-1.06), but not during the 6-9 months (OR: 1.03, 95% CI: 0.85-1.21).

Based on these additional findings, we have included a corresponding discussion in the revised manuscript. This supplementary analysis highlights the positive association between pregnancy loss and flood exposure prior to conception, providing evidence that floods may indirectly impact women's reproductive ability and well-being by disrupting essential services and affecting the social and economic basis over a prolonged period.

Please see lines 412-422 in the methods section and lines 137-142 in the results section.

12. Stratification analysis: Miscarriage (< 5 months of gestation) can be further classified as embryonic loss (or early miscarriage) when it occurs before 10 gestational weeks and fetal loss (or fetal miscarriage) when it occurs after 10 gestational weeks. Would floods have different effect on embryonic loss and fetal loss?

Response:

Thanks for this valuable comment regarding the stratification analysis of miscarriage based on gestational weeks. As miscarriages were recorded on a monthly basis, we added a stratified analysis by dividing gestational months into two subgroups (≤ 2 months; > 2 and < 5 months). We found a significant association between flood exposure and miscarriage in the group of > 2 and < 5 months (OR: 1.12, 95% CI: 1.04-1.21), but not in the group of ≤ 2 months (OR: 1.03, 95% CI: 0.96-1.11).

The weak impact in the group of miscarriages occurring during the early stages of pregnancy is more likely attributed to issues related to embryo implantation and early development, than external factors like flood exposure.

Please see lines 134 to 136 in the results section, and lines 257 to 259 in the discussion part.

13. Results: the summary statistics of flood exposure is missing.

Response:

Added accordingly (lines 121-126 and Table S3).

14. Results: the authors didn't cite Table 1 in text and the description of Table 1 is missing.

Response:

Related description of Table 1 has been added (lines 114-118).

15. Results and discussion: I suggest not talking about the strength of the relationship between urban and rural, as based on the size of the effect ORs are similar. The authors ought to present the statistical tests and display the p-value indicating the difference in odds ratios between subgroups.

Response:

As suggested, the related comparison between urban and rural has been omitted, as the difference between these two subgroups is not significant.

16. Discussion: what is the rationale behind the authors describing individuals under 25 years of age as having a non-optimal age for pregnancy?

Response:

According to this comment and the comments from Reviewer 2, in this stratified analysis, we defined the following age groups: under 21, 21-25, 25-35, and over 35 years, which is also consistent with some previous studies (1,2,3).

Our analysis indicates that floods have a significant impact among pregnant women who are under 21 or above 35 years old, but not among pregnant women aged 21-25 or 25-35. For more detailed information, please refer to lines 426-429 in the methods section, lines 147-151 in the results section, and lines 247-255 in the discussion section.

Ref:

1. Cavazos-Rehg, P.A., Krauss, M.J., Spitznagel, E.L., Bommarito, K., Madden, T., Olsen, M.A., Subramaniam, H., Peipert, J.F. and Bierut, L.J., 2015. Maternal age and risk of labor and delivery complications. *Maternal and child health journal*, 19, pp.1202-1211.
2. Schummers, L., Hutcheon, J.A., Hacker, M.R., VanderWeele, T.J., Williams, P.L., McElrath, T.A. and Hernandez-Diaz, S., 2018. Absolute risks of obstetric outcomes risks by maternal age at first birth: A population-based cohort. *Epidemiology*, 29(3), p.379.
3. de Vienne, C.M., Creveuil, C. and Dreyfus, M., 2009. Does young maternal age increase the risk of adverse obstetric, fetal and neonatal outcomes: a cohort study. *European Journal of Obstetrics & Gynecology and Reproductive Biology*, 147(2), pp.151-156.

17. Discussion: can the authors provide a more detailed explanation for the higher odds ratios observed among pregnant women below the age of 25 or above 29, particularly during the second trimester?

Response:

Thanks for noting these interesting results. We have added more detailed explanations (lines 247 to 255 or see below text).

“For women younger than 21, it is well-documented that they may have an increased risk of pregnancy complications because the reproductive system is not fully developed, making it more sensitive to external stressors such as floods. Additionally, younger women may have limited access to healthcare services and resources, which may further contribute to their vulnerability during flood events (1). For women older than 35, older age is associated with a decline in fertility and an increased risk of chromosomal abnormalities in embryos, which may lead to miscarriage or stillbirth. Moreover, older women are more likely to have pre-existing health disorders, which can be exacerbated by flood-related stressors (2).”

For the significant risk among pregnant women in the mid and late-pregnancy period, we think it can be attributed to the progressively challenging mobility and relocation options for expectant mothers in the event of potential floods.

Ref:

1. Soomar, S.M., Arefin, A. and Soomar, S.M., 2023. “Women are again unsafe”: Preventing violence and poor maternal outcomes during current floods in Pakistan. *Journal of global health*, 13.
2. Cavazos-Rehg, P.A., Krauss, M.J., Spitznagel, E.L., Bommarito, K., Madden, T., Olsen, M.A., Subramaniam, H., Peipert, J.F. and Bierut, L.J., 2015. Maternal age and risk of labor and delivery complications. *Maternal and child health journal*, 19, pp.1202-1211.

Minor comments:

1. Introduction: “The health effects of flood exposure have been well-documented for the general population, including injuries, communicable diseases, and vector-borne diseases.” The health effects of flood exposure on the general population are actually not well-documented, where the outcomes are limited to psychological disorders, infectious disease, and gastrointestinal diseases.

Response:

We've revised this part of the introduction, accordingly, please see lines 54-58 or below:

“Previous studies have reported the impacts of flood exposure on some specific diseases among the general population, such as psychological disorders, infectious diseases, and gastrointestinal diseases. There is a notable gap in the study when it comes to the significant effects of such exposure on vulnerable groups, particularly women of childbearing age have not been thoroughly investigated.”

2. Introduction: The first paragraph mainly focuses on the indirect health outcomes, and the indirect impacts on vulnerable groups. The author should establish the plausibility of the association between flood and pregnancy loss. Here is a study may be helpful: Guo, C., Chen, G., He, P., Zhang, L., & Zheng, X. (2020). Risk of cognitive impairment in children after maternal exposure to the 1998 Yangtze River flood during pregnancy: analysis of data from China's second National Sample Survey on Disability. *The Lancet. Planetary health*, 4(11), e522–e529. [https://doi.org/10.1016/S2542-5196\(20\)30198-4](https://doi.org/10.1016/S2542-5196(20)30198-4)

Response:

Following the reviewer's suggestion, we expanded our introduction to establishing the plausibility of the association between flood and pregnancy loss (see lines 58 to 63). In this section, we briefly introduced the health consequences associated with floods and their potential implications for pregnancy outcomes. Specifically, we have explicitly mentioned the direct and indirect impacts of flood exposure on pregnancy loss among vulnerable groups.

3. Introduction: “The existing epidemiological evidence for the relationship between maternal flood exposure and pregnancy loss is limited to small sample sizes, single flood events, and specific regions^{11,16}”. The references are not relevant with pregnancy loss.

Response:

We apologize for the inadequate citations and update the references accordingly (see line 68).

We have also checked all other references cited throughout the paper to ensure their relevance and appropriateness.

4. Introduction: “The existing epidemiological evidence for the relationship between maternal flood exposure and pregnancy loss is limited to small sample sizes, single flood events, and specific regions^{11,16}, making it difficult to draw reliable conclusions.” This should not be the

leading sentence of this paragraph, which focuses on research in LIMICs. The author may consider move this sentence to the last paragraph.

Response:

As suggested, we have moved this sentence to the last paragraph, please see the lines 66-68. Then we added the leading sentence of this paragraph for better connection as:

The scarcity of research conducted from a multi-region perspective hampers our understanding of the impact of flood exposure on pregnancy loss in developing countries. (lines 69-70).

5. Introduction: “and is also helpful for prioritizing approaches to deliver crucial services”. This is vague.

Response:

More comprehensive explanations have been added:

“It provides invaluable insights for effectively organizing the delivery of essential services and implementing targeted protective measures to safeguard vulnerable populations in the face of climate change. By identifying these understudied and at-risk populations, we can address their specific needs more effectively and improve their overall health outcomes.” (lines 81-86)

6. Method: “We also obtained demographic and socioeconomic information on the individual and household levels”. Is this a leading sentence of the following information? If yes, the latitude and longitude information is on a village level rather than an individual or household level. If no, please specify the individual- and household-level information.

Response:

We have made several revisions to this part. Firstly, in the mentioned sentence, we obtained all the necessary demographic and socioeconomic information at the beginning without mentioning the different levels. Then, we added more specific descriptions for other important individuals- and household-level information, including maternal age, wealth, education, type of residential area (urban or rural), water source, and floor material of the living building. Please see lines 332-336.

7. The authors mentioned 43 countries in the method section, whereas the title states 33 countries.

Response:

We initially collected data from 43 countries, but only 33 of them could be matched with flood data. We have added an explanation about this issue in the results section. Please see the lines 109-110.

8. Method: “We did not utilize this database because satellite imagery could not detect some flood events due to persistent cloud cover”. I guess the author would like to say “we did not utilize this database for the primary analysis”.

Response:

We’ve revised this sentence as “we did not utilize this database for the primary analysis”, please see line 363.

9. Analysis: “we defined the gestational flood exposure (binary variable) when the spatial location of the residing cluster during the pregnancy period was overlapped with the affected area by a flood event during the flooding period”. This has been defined in the section flood exposure data.

Response:

Based on the suggestion, we have deleted the repeated description in the section of flood exposure data.

10. Analysis: Please define “pregnancy periods”, which was used in the stratified analyses.

Response:

The pregnancy periods included early pregnancy (<4 month), mid-pregnancy (4-7 month), and late pregnancy (>7 month). Please see lines 430-431.

11. Table 1: the subheading “Floor material” should not include “the mother's age at birth”.

Response:

Following this suggestion, we have added a separate column for “the mother's age at birth” in Table 1.

12. The authors should rephrase “birth month” and “birth year” in Fig S2 in another way? The term "birth" should not be used where the outcome is pregnancy loss.

Response:

We’ve revised these as “month” and “year”.

13. Fig S2: the title of Fig S2 “different sets of controls”, should it be “different sets of covariates”?

Response:

We've revised it accordingly.

14. Fig S3: what is the meaning of the dashed line?

Response:

The dashed line in Fig S3 represents the estimated odds ratio overall study regions. The related explanation has been added.

15. Does the wealth presented in Figure 3 correspond to the income level in section “Pregnancy loss data”?

Response:

Yes, the wealth presented in Figure 3 corresponds to the income level mentioned in the "Pregnancy loss data" section. We have revised this figure to ensure consistency in our terminology.

16. The floor material was categorized as natural, unfinished, and finished in Table 1, but as natural, rudimentary, and finished in text. Please ensure consistency of the terminology used.

Response:

Thank you for pointing out the inconsistency in the terminology used to describe the floor material. We have thoroughly revised the text to ensure consistency, and now consistently refer to the floor material as "natural, rudimentary, and finished floor" throughout the manuscript.

Reviewer #2 (Remarks to the Author):

The authors performed a matched case-control study in an attempt to correlate an increased risk of pregnancy loss to flooding events during pregnancy in 33 developing countries. Individual-level data of women with pregnancy loss were retrieved from multiple DHS datasets collected between 2010 and 2020 including 69,480 pregnancy losses and 194,409 control with successful deliveries. Information about flooding events was primarily collected from the Dartmouth Flood Observatory Database. An overall odds ratio (OR) of 1.06 (95% CI 1.02-1.10) for pregnancy loss in flood-exposed pregnancies was found. Women with low income and poor education living in urban areas with an unfinished floor were in a particular high risk of pregnancy loss. The risk of pregnancy loss was only significantly increased between gestational month 4-7 and only women younger than 25 and older than 29 years had an increased risk of pregnancy loss.

1. The study is interesting and original, but the validity of the results is highly dependent on the quality of the DHS data and the possibility to correlate data from the Flooding Database with the actual flooding exposure during the women's pregnancies. I am not able to assess the quality of the data from the two databases.

Response:

We appreciate the reviewer's recognition of the study's originality and importance. We also acknowledge that the validity of our results is dependent on the quality of the data from the Demographic and Health Surveys (DHS) and the correlation between the Flooding Database and actual flooding exposure during women's pregnancies. To address this concern, we have provided additional information regarding the data sources and the implemented rigorous quality control measures on these datasets.

Firstly, the DHS datasets used in our study are widely recognized as a reliable and high-quality source of population-based data on reproductive health in developing countries. These surveys employ rigorous sampling techniques and standardized data collection protocols, ensuring a high level of data quality. Related quality control report has confirmed that through a combination of careful questionnaire design, training and supervision of interviewers, and checks and forced consistencies as part of data processing, DHS data on maternal and child health are generally of very high quality¹. Furthermore, this dataset has already been adopted by many previous studies on the topics of environmental exposure and health of pregnant

women and children^{2,3,4,5}. We have cited these studies in the methods sections to support the quality of this dataset (line 315).

In addition to the robustness of the DHS data, we have taken additional steps to verify the reliability of the records used in our study. Specifically, the health outcomes, such as pregnancy loss and its timing, were determined based on the events recorded in the reproductive calendar, which contains birth information for each respondent. Furthermore, these records were validated through a separate questionnaire that included detailed questions about pregnancy termination and duration (including questions about whether a pregnancy had been terminated, when it happened, how many months it lasted, etc.).

Regarding the Dartmouth Flood Observatory (DFO) Database, it utilizes a compilation of media reports and remote sensing imagery from reputable sources including NASA, the Japanese Space Agency, and the European Space Agency to document and analyze flood events. The flood records in the DFO can also be cross verified using the Floodlist dataset (<https://floodlist.com/>). The credibility and application of the DFO Database have been demonstrated in previous studies ^{6,7,8}.

To further strengthen our results, we conducted a sensitivity analysis using the Global Flood Database (GFD) dataset as an alternative source of flood exposure. The GFD dataset is also widely used in flood research ⁹ (<https://global-flood-database.cloudtostreet.ai/>), providing an additional layer of validation to our findings.

Overall, we have made significant efforts to ensure the validity and reliability of our study results by utilizing well-established and widely adopted data sources, employing robust validation procedures, and exploring alternative databases for corroborative analyses.

Ref:

1. Pullum TW. An Assessment of the Quality of Data on Health and Nutrition in the DHS Surveys, 1993-2003. Macro International Incorporated; 2008.
2. Xue T, Geng G, Li J, Han Y, Guo Q, Kelly FJ, Wooster MJ, Wang H, Jiangtulu B, Duan X, Wang B. Associations between exposure to landscape fire smoke and child mortality in low-income and middle-income countries: a matched case-control study. *The Lancet Planetary Health*. 2021 Sep 1;5(9):e588-98.
3. Xue T, Tong M, Li J, Wang R, Guan T, Li J, Li P, Liu H, Lu H, Li Y, Zhu T. Estimation of stillbirths attributable to ambient fine particles in 137 countries. *Nature Communications*. 2022 Nov 29;13(1):6950.

4. Pullabhotla, H.K., Zahid, M., Heft-Neal, S., Rathi, V. and Burke, M., 2023. Global biomass fires and infant mortality. *Proceedings of the National Academy of Sciences*, 120(23), p.e2218210120.
5. Wagner Z, Heft-Neal S, Bhutta ZA, Black RE, Burke M, Bendavid E. Armed conflict and child mortality in Africa: a geospatial analysis. *The Lancet*. 2018 Sep 8;392(10150):857-65.
6. Hu P, Zhang Q, Shi P, Chen B, Fang J. Flood-induced mortality across the globe: Spatiotemporal pattern and influencing factors. *Science of the Total Environment*. 2018 Dec 1;643:171-82.
7. Langlois BK, Marsh E, Stotland T, Simpson RB, Berry K, Carroll DA, Ismanto A, Koch M, Naumova EN. Usability of existing global and national data for flood related vulnerability assessment in Indonesia. *Science of The Total Environment*. 2023 May 15;873:162315.
8. Gao W, Shen Q, Zhou Y, Li X. Analysis of flood inundation in ungauged basins based on multi-source remote sensing data. *Environmental monitoring and assessment*. 2018 Mar;190:1-3.
9. Tellman B, Sullivan JA, Kuhn C, Kettner AJ, Doyle CS, Brakenridge GR, Erickson TA, Slayback DA. Satellite imaging reveals increased proportion of population exposed to floods. *Nature*. 2021 Aug 5;596(7870):80-6.

2. The information about pregnancy losses that have happened between gestational month 4-7 must be considered valid since the registration of late pregnancy losses (in contrast to earlier losses) is probably complete also in developing countries.

Response:

As we responded to the above comment, we've further verified the reliability of each of the records. The data on pregnancy loss and its incidence time was firstly determined by the events recorded in the reproductive calendar, which contains the monthly birth information for each of the respondents. Furthermore, these records would be validated by a separate questionnaire (including questions about whether a pregnancy had been terminated when it happened, how many months it lasted, etc.).

3. There is some confusing information regarding the selection of cases in the results section: 35,181 cases were identified with at least one successful pregnancy before or after the

pregnancy loss. Why should cases have given birth in addition to having experienced pregnancy loss? It is also unclear what is the 3.72 months interval between matched pregnancies in the case and control group. Is it the interval between date of pregnancy loss in cases versus date of delivery in the control group? I would like inclusion of a figure illustrating how cases and controls were identified and matched.

Response:

Firstly, we apologize for the confusion caused by our wording. The inclusion of cases with both pregnancy loss and successful delivery was according to the matched case-control design we used, which intended to enable a comprehensive analysis of multiple pregnancy outcomes within the same woman. By comparing different pregnancy outcomes from the same individual, we can better understand the potential association between flood exposure and pregnancy loss. Then, the 3.72 months referred to the average duration from the month of conception to pregnancy loss for each pregnancy loss case. We've revised this sentence accordingly.

In addition, to better explain how cases and controls were identified and matched, we provide an example in Fig S6. As depicted in Fig S6, a woman has three pregnancy records. Both the first and third pregnancies resulted in full-term vaginal deliveries, while the second pregnancy ended in a miscarriage occurring during the sixth month of gestation. Consequently, we adopted a 6-month time window, starting from the beginning of each pregnancy month, to determine whether these three pregnancy cycles were exposed to floods.

Furthermore, we have incorporated explanations in the relevant sections of the results (lines 105-108) and methods (lines 382 to 388) to provide additional clarity on the selection of cases, the interval between pregnancies, and the matching process. We expect these revisions will enhance the understanding of our study design and address these concerns.

4. I would like a discussion about the results shown in fig 2 that only women with age < 25 or > 29 years are in increased risk of pregnancy loss when exposed to flooding. Could this finding be an artefact and be due to a methodological error? I would like a further subdivision according to age e.g. < 21, 21-25, 25-29, 29-33 and 33-37 years. A more detailed subdivision will clarify which age groups are at a particular increased risk of pregnancy loss.

Response:

We don't think these findings could be due to some artifact methodological errors. To address this concern, we made further subdivisions of age groups.

Firstly, we examined the association among women of several age groups, including < 21, 21-25, 25-35, and >35 years, as women younger than 21 or older than 35 are at increased risk

for adverse pregnancy and birth outcomes compared with women of optimal childbearing age^{1,2,3}. Our findings indicate that women below the age of 21 or above the age of 35 showed a significantly higher risk of pregnancy loss (age <21: OR: 1.12, 95% CI: 1.02-1.24; age >35: OR: 1.17, 95% CI: 1.07-1.28). However, we did not observe a significant increase in risk for women aged 21-25 (OR: 1.11, 95% CI: 0.98-1.26) or 25-35 (OR: 1.02, 95% CI: 0.95-1.09).

Secondly, we expanded the discussion about the vulnerability for women of pregnancy age <21 or >35. For women younger than 21, it is well-documented that they may have an increased risk of pregnancy complications because the reproductive system is not fully developed, making it more sensitive to external stressors such as floods. Additionally, younger women may have limited access to healthcare services and resources, which can further contribute to their vulnerability during flood events. For women older than 35, advanced maternal age is associated with a decline in fertility and an increased risk of chromosomal abnormalities in embryos, which can lead to miscarriage or stillbirth. Moreover, older women are more likely to have pre-existing health conditions, which can be exacerbated by flood-related stressors.

Please refer to lines 426-429 in the methods section, lines 147-151 in the results section, and lines 247-255 in the discussion section.

Ref:

1. Cavazos-Rehg PA, Krauss MJ, Spitznagel EL, Bommarito K, Madden T, Olsen MA, Subramaniam H, Peipert JF, Bierut LJ. Maternal age and risk of labor and delivery complications. *Maternal and child health journal*. 2015 Jun;19:1202-11.
2. Schummers L, Hutcheon JA, Hacker MR, VanderWeele TJ, Williams PL, McElrath TA, Hernandez-Diaz S. Absolute risks of obstetric outcomes risks by maternal age at first birth: A population-based cohort. *Epidemiology (Cambridge, Mass.)*. 2018 May;29(3):379.
3. de Vienne CM, Creveuil C, Dreyfus M. Does young maternal age increase the risk of adverse obstetric, fetal and neonatal outcomes: a cohort study. *European Journal of Obstetrics & Gynecology and Reproductive Biology*. 2009 Dec 1;147(2):151-6.

Thanks for the valuable feedback, which has helped us enhance the comprehensiveness and clarity of our study.

Reviewer #3 (Remarks to the Author):

This manuscript presents an analysis of over 90,000 records of pregnancy loss from over 30 developing countries that have experienced incidents of major flooding between 2010-2020. Results reported include a 1.06 OR for pregnancy loss associated with floods occurring at the time of gestation. This study provides a unique analysis of understudied populations comparing the outcomes of women's non-flood related pregnancies with those that occurred during floods. Furthermore, the relevance of the sample (developing country residents) and topics (flooding and pregnancy) are well established.

1. My enthusiasm is mitigated, however, by a lack of clarity, especially involving the methods, and clear links between suggested mechanisms and outcomes. The use of undefined technical terms in sections prior to the methods is problematic. The independent understandability of the results section should be improved. I provide specific examples of this in the itemized list below.

Response:

Thanks for this valuable feedback. We appreciate the positive comments provided by the reviewer and acknowledge their concerns regarding the lack of clarity in certain sections, especially the methods section. In response to these concerns, we have made the following revisions:

For the methods section, we have carefully revised the methods section to ensure that all technical terms are clearly defined and explained. Specifically, we have incorporated detailed explanations of the technical terms used throughout the methods section. These revisions are helpful to enhance the independent understandability of the methods.

For the results section, to improve the independent understandability of the results section, we have made several changes. Firstly, we have added necessary explanations regarding the study design (see lines 98 to 108), providing a description of how the research was carried out. Additionally, we have provided further clarification for each isolated result, ensuring that they are presented in a manner that can be easily comprehended.

For the discussion section, to establish clear links between suggested mechanisms and the reported outcomes, we have provided a more thorough discussion on the results of stratification analysis that were previously not extensively explored (see lines 255 to 263).

2. Another general issue involves the mechanisms highlighted by the authors, some are mentioned in the introduction, but a more comprehensive list begins on line 185. Using stratified analyses the authors claim to have identified possible “mechanisms and population vulnerability for the risks of pregnancy loss induced by maternal flood exposure.” The paragraph goes on to list lower income, less education, less preparation for flood events, informal settlements, inefficient drainage, unplanned sanitation infrastructure, and unsafe water. However, there is no discussion of evidence relating these factors to stillbirth or miscarriage. Please explain how these factors might contribute to the specific outcomes analyzed in this report and provide evidence for these associations.

Response:

As suggested, we have revised this part of the discussion to further explain how these factors might contribute to the higher risks of pregnancy loss induced by maternal flood exposure. Please refer to lines 225 to 263 or the below text.

First, compared to houses with natural or finished floors, rudimentary floors are more prone to structural compromise during floods, rendering them less capable of withstanding flood-related challenges. As a result, these houses become more susceptible to flood, leading to unsafe living environments. This heightened vulnerability can exacerbate the risks faced by pregnant women, including physical injuries and exposure to hazardous materials.

Second, women with lower income and less education may form barriers in accessing adequate prenatal care, compared to women who are well-educated or financially privileged. These barriers can significantly impact the management of maternal health during pregnancy, potentially resulting in adverse outcomes such as stillbirth or miscarriage.

Third, women with access to unsafe water sources, flood-contaminated water sources, disrupted sanitation infrastructure, and hindered drainage, showed an increased risk because the conducive environment to the spread of waterborne diseases and infections, increasing the vulnerability to maternal infections and further risk for stillbirth or miscarriage.

Finally, for women younger than 21, it is well-documented that they may have an increased risk of pregnancy complications because the reproductive system is not fully developed, making it more sensitive to external stressors such as floods. For women older than 35, advanced maternal age is associated with a decline in fertility and an increased risk of chromosomal abnormalities in embryos, which can lead to miscarriage or stillbirth. Moreover, older women are more likely to have pre-existing health disorders, which can be exacerbated by flood-related stressors.

3. Related to the above issue, the authors also identify “unfinished floor” as a risk factor and as support for their hypothesis that “flood-related accidental injuries, harm, and physiological stress may be the dominant causes of pregnancy loss.” Please explain how “unfinished floor” indicates that there is a greater incidence of accidents, harm, physiological stress, etc. and how the data indicate that these are “dominant” causes, Also, please clarify the theoretical rationale for the hypothesis.

Response:

As we responded in the above comment, rudimentary floor conditions pertain to floors that are poorly constructed and lack proper sealing. Compared to houses with natural or finished floors, rudimentary floors are more prone to structural compromise during floods, rendering them less capable of withstanding flood-related challenges. As a result, these houses become more susceptible to damage, leading to unsafe living environments.

However, we acknowledge that our results could not prove the dominance of “unfinished floor” as the primary cause of pregnancy loss. Therefore, we have removed this problematic statement from our revised version.

For further clarification of this part of the results, we kindly direct the reviewer's attention to lines 230-234 of the revised manuscript.

4. Finally, I was surprised that there was no mention of psychological stress as a possible mechanism. Research has shown that pregnancy outcomes are impacted by the stress resulting from natural disasters during pregnancy (e.g., Dancause et al., 2011), including flooding (Hilmert et al., 2016; Tong et al., 2011). Many of the potential mechanisms, particularly SES-related factors, are stress-related.

Response:

We appreciate your suggestion regarding the inclusion of psychological stress as a potential mechanism to interpret our results. We agree that stress plays a significant role in pregnancy outcomes, especially in the context of natural disasters. To address this, we have incorporated psychological stress as one of the possible mechanisms in our revised manuscript. Additionally, we have included the references the reviewer suggested to support the association between stress resulting from natural disasters and its impact on pregnancy outcomes. Please see lines 223-224.

Specific Items:

1. ABSTRACT. Please clarify the following: The significance of “unfinished floor” is not clear; “some indirect floods impact pathways” is unclear; “nearly straight upward trend” is imprecise and unclear.

Response:

All these have been further clarified, please see the line from 33-36 and 38-39.

2. INTRODUCTION. Please explain what is meant by “may further decrease the resilience of pregnant women.” Is this at an individual level, a social resource availability level, or other?

Response:

It should be understood at both individual and social levels. At the individual level, limited access to healthcare and increased stress during flood events play a role; and at the social level, the lack of infrastructure and supporting systems compound the challenges faced by pregnant women during flood events.

A related explanation about this has been added, please see lines 76 to 77.

3. Please explain what is meant by “is important for global public health.” Will understanding how floods in these specific areas help us better understand the impact of flooding globally? Or does this statement mean something else? I wonder if it would be more effective to focus on the importance of better understanding these understudied, at risk populations in order to help those populations in particular.

Response:

As suggested, we further explained this importance (lines 81 to 86). This understanding can provide invaluable insights for effectively organizing the delivery of essential services and implementing targeted protective measures to safeguard vulnerable populations in the context of climate change. By identifying these understudied and at-risk populations, we can address their specific needs more effectively and improve their overall health outcomes.

4. RESULTS In general I found this section very difficult to follow when read prior to the methods. In general, technical terms need to be defined or more descriptive terms need to be used. For example,

Please define “pregnancy loss” in the results.

Please define “living cluster” or use a more descriptive term to improve clarity.

Please explain the research design used when describing the sample on lines 97-103. It is not clear what is being matched, whether it is between or within subjects, how 4.52

controls are matched if it is within-subject. Alternatively, the authors could consider moving this information to the Methods.

Response:

We apologize for the difficulty in relation to the order of the results section prior to the methods section, which is mandated by this journal.

First, to enhance the clarity and understandability of the Results section, we have taken the following steps. We have provided a brief description of the study design, outlining how the research was conducted. This clarification can be found in lines 98-108. Then, we have provided a brief description for each technical term, ensuring that they are presented in a manner that can be easily comprehended.

Second, in response to these specific requests, we have added clear definitions of "pregnancy loss" as the unfortunate outcome of terminating a pregnancy before the fetus reaches viability, encompassing both miscarriages and stillbirths. These definitions can now be found in the Results section on lines 99 to 100. Additionally, we have defined "living cluster" as the geographic location of the surveyed women's village or neighborhood. The definition for this term has been added in lines 104-105.

Lastly, as we mentioned in response to comment No.1, we have revised the entire manuscript to ensure that all technical terms are defined clearly throughout the paper. By implementing these changes, we aim to improve the overall clarity and comprehensibility of our findings.

5. Lines 140-142, please provide the proportion of pregnancy losses estimated to be flood related relative total pregnancy losses for those areas.

Response:

It should be 16 (CIs: 8-22) per 10,000 pregnant women per year across the selected 33 developing countries. We have added this part of results in the line 175 accordingly.

6. Please provide a rationale for analyzing the impact of different flood types in lines 146-148. Are these differences simply due to base rate frequencies?

Response:

The rationale for analyzing the impact of different flood types lies in the fact that they have distinct causes, and therefore require different preventive measures. We have provided a further explanation in our discussion, please see lines 273-275.

7. DISCUSSION, Please explain “randomly scrambled” (line 221)

Response:

It means the actual location coordinates of the clusters were intentionally altered by up to 2 km in urban areas and up to 5 km in rural areas. This design aims to protect the privacy and confidentiality of the participants as per the rules of DHS. However, it also could induce errors in assessing flood exposure.

This explanation has been added, please see the lines 280 to 283.

8. METHODS. Please clarify the following, “modification analyses.” Is this the same as moderation?; “reproductive calendar”; “clusters of participants”

Response:

To ensure consistency in the context, the term "modification analyses" has been revised to "stratified analyses" (see line 307).

Regarding the term "reproductive calendar", refers to the monthly birth information for each respondent. We have included this clarification in the revised manuscript, which can be found in lines 317-318.

Furthermore, the "clusters of participants" refer to the geographic locations of the villages or neighborhoods where the surveyed women resided. We have added an explanation for this term as well, please see the lines 328-329.

9. Please explain why the time period for this study was chosen when the methods says that data were available for Jan 1, 1992 to March 31, 2020 (Line 280)

Response:

We chose to focus on the nearest decade to ensure the inclusion of the most recent and relevant data on flood events and their potential impact on pregnancy outcomes. It enables us to capture up-to-date information and align our analysis with current environmental conditions. Furthermore, the availability of gridded global estimates of pregnant women is limited to the most recent years.

We have added an explanation for this, please see the lines 443-448.

10. The paragraph beginning on line 298 describes a database and then says “We did not utilize this database...” Please clarify the purpose of this paragraph.

Response:

We'd like to state that this database was not utilized for the primary analysis. but indeed, used for sensitive analysis purposes. As satellite imagery was unable to capture certain flood events due to persistent cloud cover. We have revised this, as it should be “We did not utilize this database for the primary analysis.”

Please see the revised sentences in line 363.

11. The Methods section includes definitions that clarify some of the issues in the prior sections, however it is still not clear to me if there were only within-subjects “matched” comparisons or between-subjects comparisons as well. Some of the results suggest that there were between-subject analyses used, and if this is the case, please clarify how this was done.

Response:

Our analyses were based on within-subjects "matched" comparisons by comparing different pregnancy records from the same individual.

To better illustrate our design, we have included an example in Fig S6. The figure depicts the pregnancy records of a woman who experienced three pregnancies. The first and third pregnancies resulted in full-term vaginal deliveries, while the second pregnancy ended in a miscarriage during the sixth month of gestation. In our study, we adopted a 6-month time window, starting from the beginning of each pregnancy month, to determine whether these three pregnancy cycles were exposed to floods (lines 382 to 388).

We hope that this example and the revised explanation provide a clearer understanding of our study design.

12. SUPPLEMENTARY MATERIAL, The supplementary material had similar clarity issues to those mentioned previously. For example, stating that Figure S3 “confirmed that our results are stable...” would be clearer with an explanation. Also, terms like “per grid square,” “estimated distributions of pregnancies,” “spatial patterns of estimates,” and the methods described in section 2 are unclear without more precise definitions and explanation.

Response:

As suggested, we have made several improvements to the supplementary material to address the clarity issues.

We have removed these unclear professional terms the reviewer mentioned and replaced them with more precise explanations. Furthermore, we have modified the description of the methods in Section 2 of the supplementary material. We have simplified and condensed the explanations to make them more concise and accessible.

13. Please note that the sentence concluding the paragraph in this section on line 77-79 seems to be out of order with the first sentence of the next paragraph. That is it seems like the authors “estimate the excess pregnancy losses related to flood exposure” before they “identified whether pregnancies... were exposed to flood events.”

Response:

We agree that such a statement is redundant and could lead to ambiguity. We have removed this repetitive paragraph and reorganized the content accordingly. Please see the revised section 2 in the supplementary material.

Reviewers' Comments:

Reviewer #2:

Remarks to the Author:

The study is very complex with a lot of potential possibilities for errors. It is thus difficult to assess the validity of the results and conclusions. However, I think the authors have done a good job in responding to the comments by the reviewers.

Reviewer #3:

Remarks to the Author:

Review of "Flood exposure and pregnancy loss in 33 developing countries" (NCOMMS-23-04056A).

This is a revision of a manuscript I reviewed previously. The authors have addressed the issues I raised adequately. I believe this manuscript makes a valuable contribution to the literature. I have no other major concerns.

Reviewer #4:

Remarks to the Author:

Floods are increasingly impacting populations worldwide as climate change increases. Yet little is known about health impacts of several important aspects of climate change, including flooding. The authors examine the pregnancy loss risk for women exposed to flood using 90,465 pregnancy losses from 33 developing countries, They then matched records with spatially explicit flood databases. The results suggest several important findings: Gestational flood exposure was associated with increased pregnancy loss and also showed a higher risk for women who relied on surface water or with lower income or education levels. The study is important in that just in one decade they find over 100,000 pregnancy losses attributed to flooding and they also find an upward trend in annual excess pregnancy losses with especially high rates in Central America, the Caribbean, South America, and South Asia. The geography of the results underscores the level of inequality in maternal and child health in relation to climate change broadly and floods more specifically.

The work is of high significance to the field and related fields across the sciences. It expands on the still nascent literature on health impacts of climate change. The original work supports the findings and conclusions. All notable issues in data analysis, interpretation and conclusions have been thoroughly addressed by the authors' response to 2 reviewers. The methodology now appears sound and replicable.

Reviewer #2 (Remarks to the Author):

The study is very complex with a lot of potential possibilities for errors. It is thus difficult to assess the validity of the results and conclusions. However, I think the authors have done a good job in responding to the comments by the reviewers.

Response: We appreciate the reviewers' great suggestions for improving our work. This analysis appears to be somewhat complex as we conducted many analyses to ensure the robustness of our results.

Reviewer #3 (Remarks to the Author):

Review of "Flood exposure and pregnancy loss in 33 developing countries" (NCOMMS-23-04056A). This is a revision of a manuscript I reviewed previously. The authors have addressed the issues I raised adequately. I believe this manuscript makes a valuable contribution to the literature. I have no other major concerns.

Response: Thanks for the valuable suggestions and positive comments on our manuscript.

Reviewer #4 (replacement for Reviewer #1 in previous round) (Remarks to the Author):

Floods are increasingly impacting populations worldwide as climate change increases. Yet little is known about health impacts of several important aspects of climate change, including flooding. The authors examine the pregnancy loss risk for women exposed to flood using 90,465 pregnancy losses from 33 developing countries, They then matched records with spatially explicit flood databases. The results suggest several important findings: Gestational flood exposure was associated with increased pregnancy loss and also showed a higher risk for women who relied on surface water or with lower income or education levels. The study is important in that just in one decade they find over 100,000 pregnancy losses attributed to flooding and they also find an upward trend in annual excess pregnancy losses with especially high rates in Central America, the Caribbean, South America, and South Asia. The geography of the results underscores the level of inequality in maternal and child health in relation to climate change broadly and floods more specifically.

The work is of high significance to the field and related fields across the sciences. It expands on the still nascent literature on health impacts of climate change. The original work supports the findings and conclusions. All notable issues in data analysis, interpretation and conclusions have been thoroughly addressed by the authors' response to 2 reviewers. The methodology now appears sound and replicable.

Response: We would like to extend our heartfelt thanks to the reviewers' constructive comments, which have undoubtedly enhanced the quality and impact of our paper. We are committed to making any further changes if required and look forward to the possibility of our work being published in *Nature Communications*.